# OMNI: Open-endedness via Models of human Notions of Interestingness

**Jenny Zhang**[1,2]   **Joel Lehman**[3]   **Kenneth Stanley**[4]   **Jeff Clune**[1,2,5]
[1]Department of Computer Science, University of British Columbia   [2]Vector Institute
[3]Stochastic Labs   [4]Maven   [5]Canada CIFAR AI Chair

## Abstract

Open-ended algorithms aim to learn new, interesting behaviors forever. That requires a vast environment search space, but there are thus infinitely many possible tasks. Even after filtering for tasks the current agent can learn (i.e., learning progress), countless learnable yet uninteresting tasks remain (e.g., minor variations of previously learned tasks). An Achilles Heel of open-endedness research is the inability to quantify (and thus prioritize) tasks that are not just learnable, but also *interesting* (e.g., worthwhile and novel). We propose solving this problem by *Open-endedness via Models of human Notions of Interestingness* (OMNI). The insight is that we can utilize foundation models (FMs) as a model of interestingness (MoI), because they *already* internalize human concepts of interestingness from training on vast amounts of human-generated data, where humans naturally write about what they find interesting or boring. We show that FM-based MoIs improve open-ended learning by focusing on tasks that are both learnable *and interesting*, outperforming baselines based on uniform task sampling or learning progress alone. This approach has the potential to dramatically advance the ability to intelligently select which tasks to focus on next (i.e., auto-curricula), and could be seen as AI selecting its own next task to learn, facilitating self-improving AI and AI-Generating Algorithms.[1]

## 1 Introduction

Provided that the real, significant challenges of AI safety and existential risk can be solved (Critch & Krueger, 2020; Bostrom, 2002; Turchin & Denkenberger, 2020; Ecoffet et al., 2020), there are tremendous gains to be had by creating more powerful AI or even AGI. A great hope for AI is that one day it can produce breakthroughs that fundamentally improve the human condition. These so-far uniquely human advancements and discoveries are the hallmark of civilization, from the invention of the wheel, to farming, vaccines, computers, and even rock and roll. Perhaps someday, AI could achieve such major breakthroughs automatically. What does AI need to possess to discover such new paradigms, as only humans have until now?

Much discussed in open-endedness research (Stanley et al., 2023), the ephemeral fuel behind civilization's prodigious output is the human intuition for *interestingness*. Drawing upon eons of human experience, we can sense potential even when we don't precisely know where it leads. Conventional Reinforcement Learning (RL) tools (e.g., intrinsic motivation (Aubret et al., 2019; Pathak et al., 2017; Osband et al., 2018; Colas et al., 2022; Oudeyer et al., 2007) and learning progress (Kanitscheider et al., 2021; Matiisen et al., 2019; Portelas et al., 2020; Graves et al., 2017; Kovač et al., 2022; Baranes & Oudeyer, 2013)) are so far only shadows of what such a human sense could do. However, with the rise of foundation models (FMs) (Bommasani et al., 2021), such as large language models (Radford et al., 2018), an intriguing prospect has arisen – trained on vast troves of human experience, perhaps FMs have the potential to grapple for the first time with the critical question of what is actually interesting to explore.

Open-ended learning algorithms, which could leverage such a notion of interestingness, seek to create AI agents that, like humans, continuously learn a variety of different skills within a vast, complex,

---

[1]We recommend the version on arXiv (`https://arxiv.org/abs/2306.01711`), which is slightly longer and thus able to explain things more clearly and more fully discuss the implications of this work. Project website: `https://www.jennyzhangzt.com/omni/`.

ever-changing environment. The challenge addressed by interestingness is that, in such environments, there are an infinite number of possible tasks, requiring some method to choose which tasks to try to learn next at every point in training. Handcrafting curricula for training agents in open-ended environments can be extremely challenging due to the sheer number of tasks and the need to adapt to the agent's skill level and learning progress. In pursuit of an algorithm that is applicable in any domain and enables perpetual learning, handcrafting curricula proves to be an impractical solution. Learning progress methods are a type of auto-curriculum approach that estimates which tasks are at appropriate difficulty levels for the agent to learn from (Kanitscheider et al., 2021; Matiisen et al., 2019; Portelas et al., 2020; Graves et al., 2017; Kovač et al., 2022; Baranes & Oudeyer, 2013). However, such methods can be distracted by learnable yet uninteresting tasks. For example, an agent could be bogged down indefinitely with rearranging silverware in slightly new configurations, hindering it from trying other interesting tasks. Even after filtering for tasks that the current agent can learn, countless learnable yet uninteresting tasks may persist (e.g., slight variations of previously learned tasks). A key challenge in open-endedness research is the inability to quantify and thus focus on tasks that are not only learnable but also interesting. There have been many attempts to quantify interestingness, but, as we detail in Section 2, such simple, hand-crafted formulas consistently fall short of truly capturing the essence of interestingness, creating crippling pathologies. This paper proposes a different path forward.

To borrow from Newton, modern AI sees further by standing on the shoulders of giant human datasets. Training on vast amounts of human-generated data has proven powerful in many cases, such as text generation (e.g., GPT-3 (Brown et al., 2020)), image generation (e.g., DALL-E (Ramesh et al., 2021)), and representation learning (e.g., CLIP (Radford et al., 2021)). We propose *Open-endedness via Models of human Notions of Interestingness* (OMNI). OMNI leverages the power of FMs that have already been trained on extensive human-generated data and have an inherent understanding of human notions of interestingness (Brown et al., 2020; OpenAI, 2023). OMNI utilizes FMs as a model of interestingness (MoI) to focus on tasks that are: (1) learnable, at appropriate difficulty levels for the agents to learn from, and (2) interesting, roughly meaning worthwhile to learn and sufficiently novel. The concepts of "interestingness", "worthwhile", and "novelty" are challenging to explicitly define, let alone quantify, which is precisely what OMNI addresses. Humans can intuitively assess these qualities despite their elusive and abstract nature, echoing Justice Potter Stewart's sentiment of "I know it when I see it" (Stewart, 1964). The goal of OMNI is to emulate this human capacity for nuanced interestingness judgement in open-ended learning. We evaluate OMNI on three challenging domains, *Crafter* (Hafner, 2021) (a 2D version of Minecraft), *BabyAI* (Chevalier-Boisvert et al., 2018) (a 2D grid world for grounded language learning), and *AI2-THOR* (Kolve et al., 2017) (a 3D photo-realistic embodied robotics environment). OMNI outperforms baselines based on uniform task sampling or learning progress alone. Overall, OMNI has the potential to significantly enhance the ability of AI to intelligently select which tasks to concentrate on next for endless learning and marks a step towards self-improving AI and AI-Generating Algorithms (Clune, 2019).

## 2 RELATED WORK

### 2.1 AUTO-CURRICULUM LEARNING

Training neural networks with a curriculum has been extensively studied (Bengio et al., 2009). Auto-curriculum learning has emerged as a promising research area in RL (Kanitscheider et al., 2021; Matiisen et al., 2019; Portelas et al., 2020; Graves et al., 2017; Kovač et al., 2022; Baranes & Oudeyer, 2013; Lehman & Stanley, 2011a; Eysenbach et al., 2018; Wang et al., 2019; 2020; Akkaya et al., 2019; Florensa et al., 2018; Zhang et al., 2020; Campero et al., 2020; OpenAI et al., 2021; Dennis et al., 2020; Gur et al., 2021; Jiang et al., 2021; Dharna et al., 2022), with approaches based on success probabilities and reward thresholds (Wang et al., 2019; 2020; Akkaya et al., 2019; Campero et al., 2020; Tan et al., 2023), regret (Dennis et al., 2020; Gur et al., 2021; Jiang et al., 2021), or learning progress (Kanitscheider et al., 2021; Matiisen et al., 2019; Portelas et al., 2020; Graves et al., 2017; Kovač et al., 2022; Baranes & Oudeyer, 2013). Static threshold-based approaches provide a straightforward method for curriculum design. These approaches involve setting fixed criteria for tasks based on their difficulty or complexity. An agent progresses to the subsequent task in a predefined order only after mastering a simpler one. To handcraft an effective curriculum, one would have to understand the relative difficulty of each task and identify tasks of suitable difficulty corresponding to each phase of the agent's learning trajectory. Doing this in a vast task space is extremely difficult or even impossible. Regret-based methods compute per-task regret by

taking the difference between the maximum known return and the average return over multiple rollouts. Regret-based methods typically select tasks with high regret, under the assumption that these tasks still offer substantial learning opportunities (Dennis et al., 2020; Gur et al., 2021; Jiang et al., 2021). However, in stochastic environments, this approach may favor more stochastic and less learnable tasks instead of less stochastic and more learnable ones (Kanitscheider et al., 2021). Learning-progress-based curricula have the potential to mitigate these issues by monitoring the agent's progress and adapting the task selection accordingly (Kanitscheider et al., 2021; Matiisen et al., 2019; Portelas et al., 2020; Graves et al., 2017; Kovač et al., 2022; Baranes & Oudeyer, 2013). Kanitscheider et al. (2021) demonstrated that learning progress can be measured reliably and that learning-progress-based curricula can be applied to hard RL problems at scale. Our work extends the learning-progress-based curriculum proposed by Kanitscheider et al. (2021). A notable limitation of existing auto-curricula approaches is their inability to distinguish between interesting and uninteresting tasks. Despite filtering for learnable tasks, open-ended environments may still contain infinite learnable but uninteresting tasks. This paper proposes a novel method for identifying and filtering interesting tasks and integrates it with a learning-progress-based auto-curriculum.

## 2.2 Attempts to Quantify Interestingness

Many prior research papers have tried to encourage a predefined metric of novelty, diversity, exploration, or open-endedness, but doing so requires quantifying these ineffable qualities. The problem is that optimizing these quantitative measures often leads to undesirable or pathological outcomes, resulting in an output that conforms to the defined metrics, rather than achieving the intended goal (Aubret et al., 2019; Pathak et al., 2017; Osband et al., 2018; Colas et al., 2022; Oudeyer et al., 2007; Etcheverry et al., 2020; Lehman & Stanley, 2011a; Mouret, 2011; Mouret & Clune, 2015; Lehman & Stanley, 2011b; Eysenbach et al., 2018; Bellemare et al., 2016; Ecoffet et al., 2019; 2021; Mendonca et al., 2023; Lehman & Stanley, 2012; Lehman et al., 2020; Nguyen et al., 2015; Auerbach & Bongard, 2010; Zhou et al., 2023; Cai et al., 2023). As Goodhart's law posits, "when a measure becomes a target, it ceases to be a good measure" (Strathern, 1997). For example, an agent might exploit a novelty measure by generating many superficially different but ultimately trivial solutions, thus undermining the goal of discovering genuinely interesting outcomes (Lehman & Stanley, 2011a). Similarly, based on how intrinsic motivation is measured, an agent could be biased towards certain types of solutions, leading to a narrow exploration of the problem space rather than developing diverse and valuable insights and innovations (Aubret et al., 2019). Attempting to manually specify a criteria for what constitutes an interesting learning challenge is unlikely to yield satisfactory results. Instead, this paper proposes harnessing FMs to model ineffable human notions of interestingness, gleaned from large text corpora of existing human-generated data (e.g. training on the Internet).

## 2.3 Pre-trained Foundation Models in Open-Endedness

Large language models have recently shown a remarkable ability to capture rich knowledge on an extensive array of subjects from large-scale text corpora. They achieve impressive performance across a wide range of natural language processing tasks (Brown et al., 2020; OpenAI, 2023; Kenton & Toutanova, 2019; Liu et al., 2019; Min et al., 2021; Li et al., 2022; Colas et al., 2023) and display profound understanding of complex concepts such as physics. Consequently, they are utilized in many robotics domains (Huang et al., 2022b;a; Ahn et al., 2022; Yang et al., 2023; Lynch & Sermanet, 2020; Sharma et al., 2021; Kant et al., 2022; Kwon et al., 2023; Du et al., 2023; Driess et al., 2023). There has been growing interest in using them for task selection or generation. Some studies have investigated the application of FMs in breaking down high-level instructions into a sequence of sub-goals, which can be executed by an agent in a zero-shot manner (Huang et al., 2022b;a; Ahn et al., 2022; Yang et al., 2023; Colas et al., 2023; Zhu et al., 2023) or used to train modular sub-policies (Lynch & Sermanet, 2020; Sharma et al., 2021). Kant et al. (2022) queries FMs for zero-shot commonsense priors and apply them to a planning task. Other studies have utilized FMs to estimate success rates for a given task or desired behavior (Kwon et al., 2023; Du et al., 2023; Colas et al., 2023; Wang et al., 2023a;b). Moreover, FMs have been employed to generate or explain tasks, enabling structured exploration in various environments (Du et al., 2023; Colas et al., 2023; Wang et al., 2023a;b; Yuan et al., 2023). OMNI differs from Du et al. (2023) by considering an agent's past successes and employing FMs' commonsense knowledge for adaptive task selection. Unlike Wang et al. (2023a), which employs a code API generated by FMs, OMNI promotes direct action learning via environment interaction, demanding potentially higher computational resources but bypassing the need for and, critically, limitations of, domain-specific code APIs. While Colas et al. (2023) use

deterministic environments and binary reward signals for trajectory success, OMNI adopts a more nuanced approach in stochastic settings, recognizing that agents often improve over time and may not always achieve consistent success rates.

## 3 METHODS

### 3.1 PROBLEM FORMULATION

We train task-conditioned agents, and formulate the RL problem as a partially observed Markov decision process (Kaelbling et al., 1998) defined by a tuple $(\mathcal{S}, \mathcal{A}, \mathcal{T}, \mathcal{R}, \mathcal{O}, \Omega, \gamma)$. Observations $o \in \Omega$ depend on the new environment states $s \in \mathcal{S}$ and actions taken $a \in A$ via $\mathcal{O}(o|s, a)$. The task which the agent is conditioned on is part of the environment state $s$. $\mathcal{T}(s'|s, a)$ describes the dynamics of the environment. $\mathcal{R}(s, a)$ is the environment's reward function. $\gamma$ is a discount factor. OMNI focuses on generating learnable and interesting tasks to condition the RL agent on.

### 3.2 LEARNING PROGRESS CURRICULUM

The task pool in open-ended environments can be very large and diverse, making it challenging for an agent to learn effectively through uniform sampling. Most randomly sampled tasks are likely to be impossible (or at least currently too hard for the agent to learn). To automatically identify tasks at the frontier of the agent's capabilities, we extend the learning-progress-based curriculum (without the dynamic exploration bonus) from Kanitscheider et al. (2021).

The curriculum predominantly samples tasks with high learning progress, defined as an agent's recent change in task success probability. During training, the agent is periodically evaluated, and a recent success probability estimate $p_{recent}$ is calculated by applying an exponential moving average (EMA) to the evaluated task success rates. $p_{recent}$ is smoothed with a second, identical EMA to obtain a slower-to-change reflection $p_{gradual}$ of the success probability. Since tasks with low success probabilities are more likely to be novel and are harder to learn because the agent observes fewer successes, $p_{recent}$ and $p_{gradual}$ are reweighted to magnify the learning progress in tasks with low success probabilities and reduce the learning progress in tasks with high success probabilities. This reweighting also compensates for the temporal delay caused by the EMA (Figure 4). Bidirectional learning progress, the absolute difference between the reweighted $p_{recent}$ and $p_{gradual}$, is used to also focus learning on tasks where performance is degrading due to forgetting. Sampling of training tasks is biased towards those that score the highest on this bidirectional learning progress measure. We propose an extension to the approach from Kanitscheider et al. (2021), normalizing the task success rates with the success rates achieved by a random action policy (Appendix A).

### 3.3 MODELING WHAT HUMANS FIND INTERESTING

An LP curriculum can be distracted by endless variations of uninteresting tasks. To address this challenge, a Model of Interestingness (MoI) selects interesting tasks that offer substantial learning value. Humans often intuitively know what might be useful for learning new skills or achieving goals much later (Stanley & Lehman, 2015). This is evident in children playing to unknowingly acquire skills, or scientists exploring new areas to uncover unexpected and beneficial knowledge for future endeavors. This paper presents two (of many possible) instances of the OMNI principle: one in finite task spaces (Section 3.3), and one in an infinite task space (Section 5.1). This section describes OMNI the former, first outlining the process of using an FM to determine which tasks are interesting, and then describing how the interestingness predictions are utilized to obtain task sampling weights.

**Determining Interesting Tasks.** This paper capitalizes on the capabilities of autoregressive FMs to emulate human notions of interestingness. FMs are pretrained on vast and diverse text corpora, enabling them to amass a significant amount of world knowledge. We prompt the FM in a few-shot manner by providing it with examples of choosing which tasks are interesting. It takes into account the agent's existing proficiency on a given set of tasks and suggests what humans would typically find interesting to learn next. Davinci GPT-3 (Brown et al., 2020) was utilized for the Crafter experiments because it was the state-of-the-art language model available when the experiments were run. GPT-4 (OpenAI, 2023) was used for the BabyAI experiments, which were conducted later. Appendices B and C show the full prompts.

**Sampling Weights.** OMNI aims to improve open-ended learning by focusing on tasks that are both learnable and interesting (Figure H). The full OMNI algorithm is summarized in Algorithm 1. Task sampling rates are first assigned based on the LP curriculum, with higher rates for tasks with

higher learning progress (Section 3.2). Then, an FM-based MoI predicts which tasks are interesting (Appendix I). Boring tasks have their sampling weights reduced by multiplying by 0.001. Finally, task sampling rates are normalized to probabilities that sum to 1.

# 4 EXPERIMENTS IN A FINITE TASK SPACE

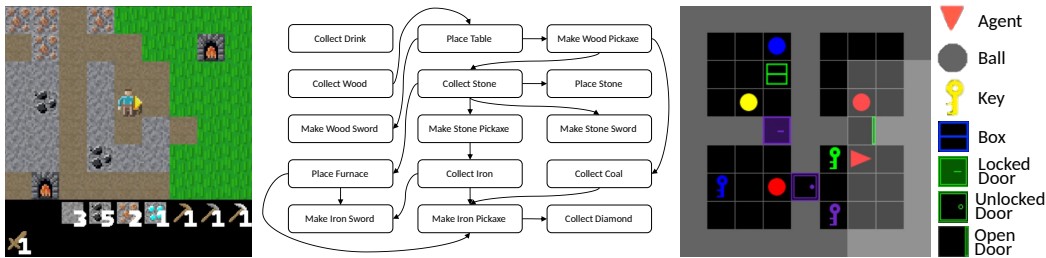

Figure 1: **Crafter and BabyAI environments. (Left)** Agent view in a procedurally generated Crafter world, showing terrain types, resources, and the agent's inventory. **(Middle)** The 15 tasks considered interesting for Crafter analyses. Arrows indicate which tasks in the technology tree must be completed, often multiple times, along the way to perform more challenging tasks. **(Right)** Bird's-eye view of a randomly generated BabyAI environment, showing different object types, colors, locations, and states. The agent is the red triangle and its view (sometimes occluded) is highlighted in light grey. In this example, the agent starts from the bottom right room, and is tasked to "go to a red ball". To succeed, the agent must open the green door (sometimes locked) to reach the red ball.

## 4.1 CRAFTER ENVIRONMENT

We evaluate OMNI on Crafter (Hafner, 2021), a 2D version of Minecraft that enables collecting and creating a set of artifacts organized along a technology tree. This means that certain tasks need to be completed, often multiple times, as prerequisites for other more challenging tasks (Figure 1). Agents receive RGB pixel observations (64 x 64 resolution) of a 9 x 9 grid area surrounding their position within a 64 x 64 grid landscape that varies with each episode, offering a complex and engaging testing ground. The agent is provided a target task (represented with a bag-of-words encoding) as part of its observation and rewarded +1 upon successful completion of the conditioned task. We modify the game to focus on gathering and crafting skills by eliminating the survival component. This removes the need for the agent to learn and continually apply survival tactics against enemies or for food gathering. The "sleep" and "place plant" actions are important for survival in the original game and have been omitted due to their reduced relevance in our modified context, which excludes the survival aspect. The original game consists of 22 tasks, of which, the 15 tasks unrelated to survival are selected and considered interesting.

To investigate our hypothesis that focusing on interesting tasks with high learning progress will improve performance, we dilute the 15 interesting tasks with 90 "boring" tasks and 1023 "extremely challenging" tasks that serve as potential distractors for learning-progress-based approaches. Boring tasks are generated as numerical repeats of interesting tasks, e.g., "collect $N$ wood" where $N \in [2, 10]$, analogous to how minor numerical variations of real-world tasks are less interesting than tasks that differ qualitatively. See Appendix K for the full list of boring tasks. Extremely challenging tasks represent tasks that are too difficult for the agent to complete at its current state of learning, serving as tasks that uniform sampling will waste time on, but that learning-progress-based methods should successfully ignore. The agent is assumed to always fail at these extremely challenging tasks and hence is always assigned a success rate of 0 for them. By analogy, consider the futility of attempting to cook a 5-course meal before learning the basic skill of cutting a vegetable.

## 4.2 BABYAI ENVIRONMENT

We also evaluate OMNI on BabyAI (Chevalier-Boisvert et al., 2018), a readily available benchmark domain characterized by its partially observable 2D grid world environment (Figure 1). We test on the *MiniBossLevel*. While *BossLevel* is the most challenging level in BabyAI, we choose *MiniBossLevel* as it has the same features as *BossLevel* but with a smaller room and lower probability of locked rooms, speeding up training. For each episode, the room layout and item configuration are randomly generated (using off-the-shelf configurations from Chevalier-Boisvert et al. (2018)). The grid world

can have objects in six colors (*red, green, blue, purple, yellow, grey*), and of four types (*key, ball, box, door*). The agent is randomly spawned at a location in the 9 x 9 grid world, containing four 3 x 3 rooms. The agent's observation includes one-hot encodings of each of the 7 x 7 grid cells in front of the agent (observations are set to a special symbol if occluded), and a description of the task in natural language (embedded with a look-up table and GRU, see Appendix M for more details). The agent receives a reward, proportional to the number of steps it took to finish, only when it has successfully completed the given task. While the Baby Language grammar (Chevalier-Boisvert et al., 2018) is limited to sequential tasks with a maximum of 2 instructions, we expanded this by permitting tasks with up to 5 instructions, resulting in 1364 unique tasks. Each task is a sequence of instructions (*GoTo, PickUp, OpenDoor, PutNextTo*), linked by the ordering constraint *then*. Object placements are randomized each episode. Tasks with the same sequence of instructions but different object instances are considered the same when sampling (e.g., "go to a blue ball" and "go to a red key" are considered the same task "go to <object>").

## 4.3 RESULTS

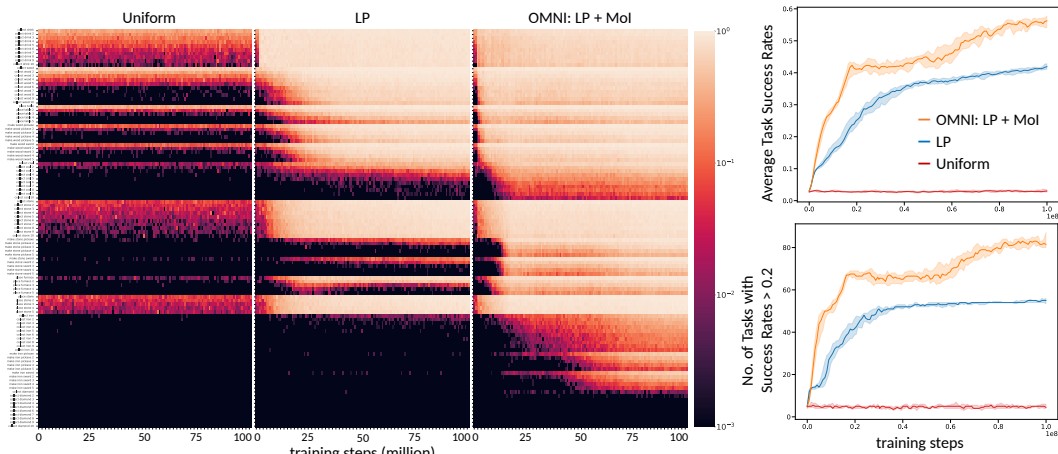

Figure 2: **Results in Crafter.** (**Left**) Conditional success probabilities of all tasks in Crafter. Tasks are organized from simple to complex based on the prerequisite tasks that must be accomplished before completing the target task. Task names (left of each row) are readable in a digital format with zoom. (**Right**) Performance in Crafter on all tasks. While OMNI biases training towards *interesting tasks*, it achieves higher average task success rates and learns more tasks than uniform sampling or choosing tasks based on learning progress alone, even across *all tasks*.

Both Crafter and BabyAI RL agents are trained with PPO (Schulman et al., 2017), a standard RL algorithm. Policy details and hyperparameters for the Crafter and BabyAI settings are in Appendices L and M. We compare the performance of agents trained with: (1) **Uniform** sampling, (2) Learning Progress (**LP**) only, and (3) OMNI: Learning Progress with additional filtering by a Model of Interestingness (**OMNI: LP + MoI**). Uniform sampling, the control, samples all tasks with equal probabilities. Uniform sampling is the most naive and samples tasks that are too easy or too difficult for the agent most of the time. LP samples tasks based on the calculated learning progress weights (Section 3.2), but is distracted by the many boring tasks. OMNI: LP + MoI focuses on the subset of tasks with high learning progress that are also interesting (Section 3.3). All experiments are run for 100 million time steps and are repeated 10 times with different random seeds. Each experiment takes about 33 hrs for Crafter and 60 hrs for BabyAI on a 24GB NVIDIA A10 GPU with 30 virtual CPUs.

We evaluate our methods with two metrics: (1) the average task success rate, and (2) the number of tasks with success rates exceeding a predetermined threshold $\alpha$. This study sets $\alpha = 0.2$, consistent with the selections made in related literature (Kanitscheider et al., 2021; Team et al., 2023). The first metric reflects the agent's average performance across all tasks, while the second metric captures the extent to which the agent is a generalist that has decent competency on many different tasks. These metrics are calculated on the full task set (Figures 2 and 7). Metrics calculated on interesting tasks only are shown in Appendix O. All confidence intervals given are 95% median bootstrap confidence intervals obtained by resampling 1000 times. Confidence intervals are reported with the following

notation: $stat$ (CI: $lower - upper$) where $stat$ is the median across runs. Shaded areas in graphs also indicate the 95% median bootstrap confidence interval obtained by resampling 1000 times.

**Uniform Sampling.** As expected, the results with uniform sampling are poor. Worse, the agents did not improve over time as most tasks sampled are too difficult or too easy for the agent and successes are extremely sparse (Figures 2 and 7). The agent is considered to have learned a task if its conditional success probability on that task is at least 0.2. In Crafter, the agent learns 4 (CI: $4 - 6$) tasks (interesting or boring) and only 3 (CI: $2 - 3$) interesting tasks. The agent achieves an average task success rate of 0.030 (CI: $0.026 - 0.033$) on interesting and boring tasks, and 0.103 (CI: $0.087 - 0.120$) on interesting tasks only. In BabyAI, the agent learns only 1 (CI: $0 - 1$) task and achieves an average task success rate of 4.7e-3 (CI: 4.6e-3 – 5.0e-3) on all tasks.

**Learning Progress Curriculum.** By focusing on tasks with suitable difficulty, the agent learns to do a lot more tasks with higher success rates than uniform sampling. In Crafter, the agent learns 55 (CI: $54 - 56$) tasks (interesting or boring) and 9 (CI: $9 - 11$) interesting tasks. The agent achieves an average task success rate of 0.42 (CI: $0.41 - 0.43$) on interesting and boring tasks, and 0.52 (CI: $0.50 - 0.56$) on interesting tasks only. In BabyAI, the agent learns 4 (CI: $4 - 6$) tasks and achieves an average task success rate of 5.9e-3 (CI: 5.5e-3 – 6.2e-3) on all tasks. Across all metrics and in both domains, the differences in performance between LP and Uniform at 25%, 50%, 75%, and 100% of the way through training are statistically significant (all p < 1e-3, Mann Whitney U test), showing that LP significantly outperforms uniform sampling (Figures 2 and 7). LP samples tasks that are at the frontier of the agent's capabilities (Figures 2, 7, 8, 14). When a task's conditional success probability changes, LP focuses more on it. Hence, there will be more rollouts where the task is the given goal and thus more positive examples from which the agent can learn to solve the conditioned task. However, LP is distracted by boring tasks (Figures 8 and 14). When the conditional success probabilities of boring tasks change, LP allocates higher sampling weights to them even though they are similar to other sampled tasks and might not expand the agent's range of skills.

**OMNI: Learning Progress + a Model of Interestingness.** To automatically select and focus on interesting tasks, an FM is prompted in a few-shot manner to predict which tasks are interesting. By combining LP with an MoI, OMNI focuses on the subset of high learning progress tasks that are interesting. In Crafter, the agent learns 82 (CI: $80 - 87$) tasks (interesting or boring) and 14 (CI: $14 - 14$) interesting tasks. The agent achieves an average task success rate of 0.56 (CI: $0.54 - 0.58$) on interesting and boring tasks, and 0.78 (CI: $0.76 - 0.80$) on interesting tasks only. In BabyAI, the agent learns 8 (CI: $7 - 10$) tasks and achieves an average task success rate of 7.5e-3 (CI: 7.3e-3 – 7.7e-3) on all tasks. Across all metrics and in both domains, the differences in performance between OMNI and LP at 25%, 50%, 75%, and 100% of the way through training are statistically significant (all p < 1e-3, Mann Whitney U test), showing that OMNI significantly outperforms an LP-only curriculum (Figures 2 and 7). OMNI is not distracted by uninteresting yet learnable tasks, and focuses on the interesting tasks only (Figures 8 and 14). The trained agent not only achieves higher average task success rates, but also learns more challenging tasks faster (Figures 2 and 7).

We thus know OMNI performs better than LP alone, but how good is it at predicting interesting tasks? To address this, we created an oracle for the MoI, termed the Oracle Model of Interestingness (OMoI). Impressively, the performance of the FM-based MoI is nearly on par with the oracle, suggesting that OMNI is highly effective in identifying interesting tasks for the agent to learn on (Appendix P).

## 5 EXPERIMENTS IN AN INFINITE TASK SPACE

In truly open-ended settings, there are an infinite number of possible tasks. This section demonstrates OMNI in such a setting. Essential to training an agent capable of handling any task in such an open-ended learning framework is the development of a universal reward function, which can evaluate if *any* task has been completed or not. This section proposes an instantiation of OMNI that solves that problem by not only having FMs propose new, interesting tasks, but also by having the FM generate the code for a reward function that determines to what extent each proposed task has been performed.

### 5.1 METHODS

In an infinite task space, it is impossible to evaluate every possible task to determine the agent's learning progress. Hence, instead of using a predefined set of tasks, we use a pretrained autoregressive FM, GPT-4 (OpenAI, 2023), to generate learnable and interesting tasks throughout training. The LP curriculum then produces task sampling rates over this growing task set (Section 3.2). We input tasks

that the agent can do well and tasks that the agent cannot do yet, then prompt GPT-4 in a zero-shot manner to suggest the next learnable and interesting tasks. Tasks done well are those completed with success rates greater than a predefined threshold (0.6 in AI2-THOR experiments). We also ask GPT-4 to output a sequence of environment states (in code format) that can be used to check whether or not the task has been successfully completed during training and evaluation. Appendix D shows the full prompt and Appendix E shows an example output.

There are existing approaches that use FMs to generate code as reward functions (Kwon et al., 2023; Wang et al., 2023a; Yu et al., 2023). This version of OMNI integrates the generation of the task and the code requirements for task completion into a single output. This integrated approach ensures that every generated task comes with a comprehensive definition of what constitutes its completion (in code format). This approach can work for any domain in which one can run code to make queries about the underlying state. We apply OMNI to a complex, embodied robotics kitchen domain, AI2-THOR (Kolve et al., 2017), and show that OMNI is not only able to continuously generate learnable and interesting tasks, but also learns more tasks over time than controls.

## 5.2 AI2-THOR ENVIRONMENT

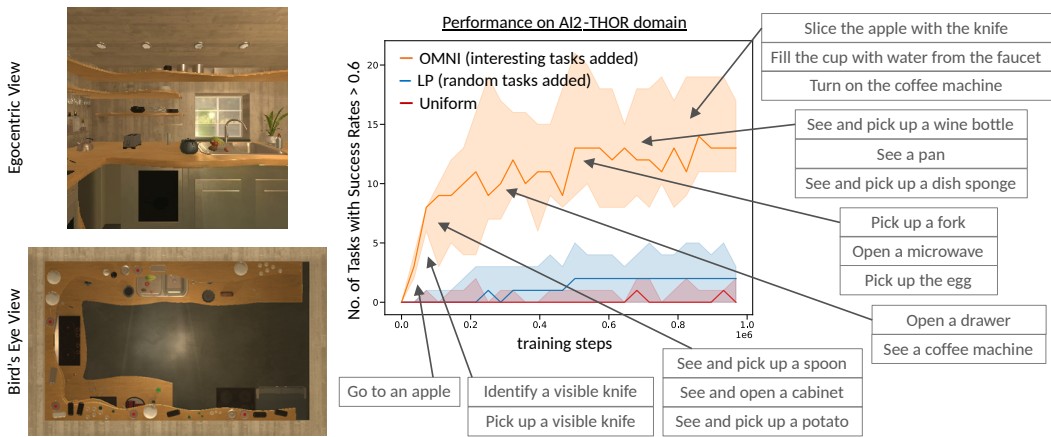

Figure 3: **AI2-THOR environment and results.** (**Left**) Agent's egocentric view and bird's-eye view in an AI2-THOR kitchen environment. (**Right**) OMNI learns more tasks than the Learning Progress and Uniform sampling baselines. Example tasks learned by OMNI are shown in gray boxes.

AI2-THOR (Kolve et al., 2017) is an embodied 3D domain characterized by its near photo-realistic environment (Figure 3). We train our methods on an AI2-THOR kitchen floorplan. The environment contains many objects commonly found in a real kitchen, such as food (e.g., apple, bread), appliances (e.g., coffee machine, microwave), and tools (e.g., mug, pan). The agent has 13 discrete actions: `MoveAhead`, `RotateRight`, `RotateLeft`, `LookUp`, `LookDown`, `Pickup`, `Put`, `Open`, `Close`, `ToggleOn`, `ToggleOff`, `Slice`, and `FillWithLiquid`. We simplify the action mechanics that require a target object as an argument (e.g., the `Pickup` action, which requires a target object like `Cup`). Rather than force the agent to specify one of an infinite number of possible objects, instead, if the object mentioned in the current task is visible and requires the action to be applied to it to complete the task, it is automatically designated as the target object. If not, the target defaults to the visible object nearest to the agent. The agent's observation includes 300 x 300 RGB pixel observations of a 90°field of view, and a description of the task in natural language (embedded with a look-up table and GRU, Appendix N). The agent receives a +1 reward, with a small penalty of 0.001 for each time step, when it has successfully completed the given task. A task can be described in natural language or by a sequence of environment states. For the agent to complete a given task, it needs to sequentially achieve a list of environment states (specified in code). For example, if the task is "Pick up an apple, then put it down", the corresponding code format could be `[[obj_attributes("Apple", "isPickedUp":  True)],` `[obj_attributes("Apple", "isPickedUp":  False)]]`, whereby the agent has to achieve the first environment state where the apple is picked up, then achieve the second environment state where the apple is not picked up. The task space is infinite, as there is no restriction on the number of attributes to check for in each environment state, or the length of environment states to be achieved sequentially when specifying each task.

The complexity and variability of tasks and interactions in AI2-THOR are significant, yet represent only a fraction of the possibilities of a *Darwin Complete* environment generator, meaning one that can create *any* possible learning environment Clune (2019). By demonstrating OMNI in this infinite AI2-THOR task space, we mark a step towards that ultimate, lofty goal of generating learnable and interesting tasks in a search space that includes any conceivable environment.

### 5.3 RESULTS

AI2-THOR RL agents are trained with PPO (Schulman et al., 2017), a standard RL algorithm. Policy details and hyperparameters are in Appendix N. We compare the performance of agents trained with: (1) **Uniform** sampling (Appendix J.1), (2) **LP**, the Learning Progress curriculum over a growing task set where random tasks are added (Appendix J.2), and (3) **OMNI**, which is the Learning Progress curriculum applied over a growing task set where interesting and learnable tasks suggested by the FM are added (Section 5.1). Uniform sampling, the control, uniformly samples any task within the task space. Uniform sampling is naive and samples tasks that are too difficult for the agent most of the time, hurting learning (before even factoring in whether the tasks are *worth* learning). LP samples tasks based on the calculated learning progress weights (Section 3.2), but most tasks added to the task set are too difficult. OMNI automatically generates learnable and interesting tasks for the agent to learn on. All experiments are run for 1 million time steps and are repeated 10 times with different random seeds. Each experiment takes ~24 hrs on a 24GB NVIDIA A10 GPU with 30 virtual CPUs.

In this vast landscape of infinite potential tasks, it is impossible to evaluate on every conceivable task. Hence, each method is only evaluated on tasks that have ever been sampled before. We measure our methods by the number of tasks completed at a success rate greater than a predetermined threshold (here, 0.6). All confidence intervals are 95% median bootstrap confidence intervals obtained by resampling 1000 times. Confidence intervals are reported with the following notation: $stat$ (CI: $lower - upper$) where $stat$ is the median across runs. Shaded areas in graphs also indicate the 95% median bootstrap confidence interval obtained by resampling 1000 times.

**Uniform Sampling.** As expected, the results with uniform sampling are poor. The agent trained with Uniform sampling learns 0 (CI: $0 - 2$) tasks (defined here and other treatments as a conditional success probability of at least 0.6).

**Learning Progress Baseline.** Although the LP curriculum allows the agent to focus on the learnable tasks within the task set, because the tasks added to the task set are often too difficult, the agent does not learn many tasks either. The agent trained with LP learns 2 (CI: $0 - 3$) tasks.

**OMNI.** To automatically generate and learn interesting tasks, an FM is prompted in a zero-shot manner to suggest the next new learnable and interesting tasks, augmenting the task set for the agent to train on. The agent trained with OMNI learns 13 (CI: $11 - 17$) tasks. The difference in performance between OMNI and both baselines (Uniform sampling and LP baseline) at 25%, 50%, 75%, and 100% of the way through training are statistically significant (all p < 1e-3, Mann Whitney U test), showing that OMNI significantly outperforms both Uniform sampling and the LP baseline (Figure 3).

## 6 DISCUSSION, FUTURE WORK, AND CONCLUSION

In conclusion, our work demonstrates the potential of using an MoI to significantly enhance auto-curricula and the quest for open-ended learning algorithms by intelligently focusing on learnable and interesting tasks. OMNI addresses the Achilles Heel of open-ended systems, which lies in defining and quantifying interestingness, as previous attempts have resulted in pathologies when optimizing against such definitions and quantifications. OMNI mitigates this problem by leveraging human notions of interestingness to guide AI systems. There are numerous ways to implement the principles of this new paradigm, and exploring different versions presents an exciting avenue for future research (Appendix U). The generality and applicability of OMNI to other open-ended domains with vast task spaces further underscores its significance. In the long run, it hints at a synergy between FMs and open-endedness that simultaneously addresses looming challenges for both: how will FMs ultimately rise to the level of creativity seen in the best of human innovation, and how will open-endedness overcome the trap of diverging into a vast space of uninspiring mediocrity? By playing off each other's strengths, FMs can perhaps someday become essential engines of open-ended discovery and begin to participate in the creative dance that has defined civilization since its inception.

ACKNOWLEDGMENTS

This work was supported by the Vector Institute, the Canada CIFAR AI Chairs program, a grant from Schmidt Futures, an NSERC Discovery Grant, and a generous donation from Rafael Cosman. We also thank Andrew Dai, Cédric Colas, and members in our lab at the University of British Columbia, namely Aaron Dharna, Ben Norman, and Shengran Hu, for insightful discussions and feedback.

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

# A    LEARNING PROGRESS CURRICULUM DETAILS

The reweighting mechanism magnifies differences in low probabilities, putting additional focus on tasks that have high learning progress and low success rates. However, it causes a bias against tasks with initially high success rates achieved by chance even though the agent has not yet learned them. We do not remove the reweighting mechanism since it remains useful in the later stages of training. Instead, we propose an extension to the approach from Kanitscheider et al. (2021), normalizing the task success rates with the success rates achieved by a random action policy. Normalizing the success rates reduces the bias against tasks with initially high success rates, increasing their sample frequency during the early stages of the learning progress curriculum.

For each task, we calculate the task success rate $t_{rdn}$ achieved by a random action policy. At fixed evaluation intervals during training, the evaluated task success rate $t_{eval}$ of the RL policy is normalized as such:

$$t_{norm} = \frac{t_{eval} - t_{rdn}}{1 - t_{rdn}}$$

The above is our contributed extension to the learning progress method in Kanitscheider et al. (2021). $t_{norm}$ is smoothed with an EMA function to obtain $p_{recent}$ (Figure 4, green). $p_{recent}$ is smoothed with a second identical EMA function to obtain $p_{gradual}$ (Figure 4, brown). The exponential smoothing constant applied in all experiments is 0.1. The bidirectional learning progress measure is given by $LP = |f(p_{recent} - f(p_{gradual})|$ (Figure 4, blue), where $f$ is the reweighting function:

$$f(p) = \frac{(1 - p_\theta)p}{p + p_\theta(1 - 2p)}$$

with parameter $p_\theta = 0.1$.

We employ a sampling function to transform the measure of learning progress into task sampling weights, focusing mostly on tasks with the largest learning progress. The steps are as follows:

- Z-score the reweighted learning progress (subtract mean and divide by standard deviation).

- Apply a sigmoid to the result.

- Normalize resulting weights to sampling probabilities.

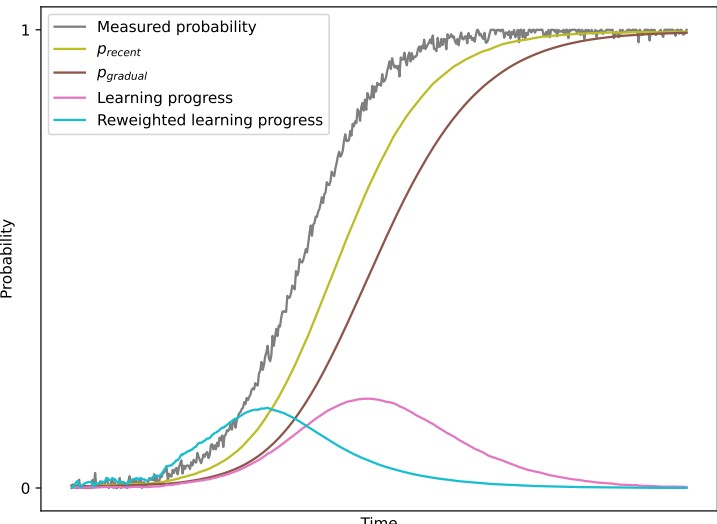

Figure 4: The process of determining an agent's learning progress on a task from its measured success probability on that task in an example fictional problem.

## A.1 LEARNING PROGRESS CURRICULUM ABLATION

The learning progress curriculum without the proposed extension (Section 3.2) of normalizing the task success rates to a random action baseline is labelled as **LP-no-norm**. In light of limited compute, we limit ourselves to the Crafter domain and experiments are run for 30 million time steps (vs. 100 million in the main experiments) and are repeated 10 times (same as the main experiments) with different random seeds. The LP curriculum achieves higher average task success rates and learns more tasks with the proposed normalization extension (Figure 5). Across all metrics and in both domains, the differences in performance between LP and LP-no-norm at 50%, 75%, and 100% of the way through training are statistically significant (all $p < 0.05$, Mann Whitney U test).

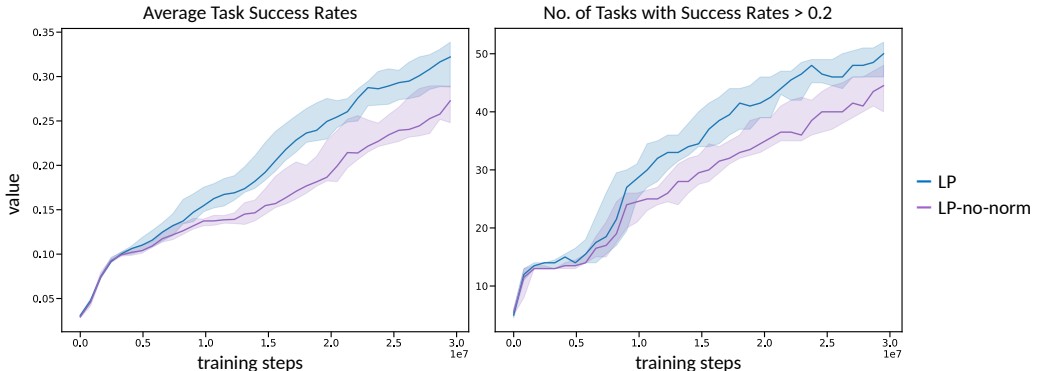

Figure 5: **Performance of learning progress curriculum in Crafter with and without the proposed normalization extension.** Average task success rates and number of tasks with success rates more than 0.2 for learning progress curriculum with and without the proposed normalization extension across training steps. Metrics are calculated on all tasks. LP performs better with the proposed normalization extension.

## B CRAFTER PROMPT

We give GPT-3 three examples as part of the prompt:

```
You are a player in a game.  You want to learn as many different
skills as possible.  You can do this task well:  collect wood.
Suggest whether the given tasks are interesting:  collect drink,
collect wood, make stone sword, make wood pickaxe, place furnace.
collect drink:  True
collect 2 drink:  True
collect 3 drink:  True
collect wood:  False
collect 2 wood:  False
collect 3 wood:  False
make stone sword:  True
make 2 stone sword:  True
make 3 stone sword:  True
make wood pickaxe:  True
make 2 wood pickaxe:  True
make 3 wood pickaxe:  True
place furnace:  True
place 2 furnace:  True
place 3 furnace:  True

You are a player in a game.  You want to learn as many different
skills as possible.  You can do this task well:  make 2 iron
pickaxe.
Suggest whether the given tasks are interesting:  collect coal,
```

```
collect iron, make iron pickaxe, make iron sword, place table.
collect coal:  True
collect 2 coal:  True
collect 3 coal:  True
collect iron:  True
collect 2 iron:  True
collect 3 iron:  True
make iron pickaxe:  False
make 2 iron pickaxe:  False
make 3 iron pickaxe:  False
make iron sword:  True
make 2 iron sword:  True
make 3 iron sword:  True
place table:  True
place 2 table:  True
place 3 table:  True

You are a player in a game.  You want to learn as many different
skills as possible.  You can do this task well:  place 3 stone.
Suggest whether the given tasks are interesting:  collect diamond,
collect stone, make stone pickaxe, make wood sword, place stone.
collect diamond:  True
collect 2 diamond:  True
collect 3 diamond:  True
collect stone:  True
collect 2 stone:  True
collect 3 stone:  True
make stone pickaxe:  True
make 2 stone pickaxe:  True
make 3 stone pickaxe:  True
make wood sword:  True
make 2 wood sword:  True
make 3 wood sword:  True
place stone:  False
place 2 stone:  False
place 3 stone:  False
```

Given the set of tasks that the agent can do relatively well and the set of tasks that needs to be determined as interesting or not, the additional prompt is:

```
You are a player in a game.  You want to learn as many different
skills as possible.  You can do these tasks well:  <tasks done
well>.
Suggest whether the given tasks are interesting:  <tasks to be
determined>.
```

In the few-shot examples, we do not include all possible tasks in `<tasks to be determined>` to reduce token usage. While each example sets different tasks for `<tasks to be determined>`, during inference, all tasks needing classification as interesting or not (Section 3.3) are inputted as `<tasks to be determined>` without any additional filtering.

GPT-3 predicts whether each task in `<tasks to be determined>` is interesting or not with `True`/`False`, following the format in the few-shot examples. In our experience, GPT-3 nearly always conforms to the requested output format. However, in the rare cases that GPT-3 deviates from the expected output format, the responses are regenerated. For tasks where GPT-3 did not provide an answer, we modify the `<tasks to be determined>` input to include only the tasks lacking responses, and then regenerate these responses.

We access GPT-3 through OpenAI's APIs, opting for the Davinci model in our experiments, which costs $0.02 per 1000 tokens when these experiments were run. Caching significantly reduces the

number of API queries. We extensively cache GPT-3 prompts and responses and consistently reuse this cache across multiple runs.

## C BABYAI PROMPT

Full system prompt to GPT-4:

```
You are a helpful assistant that tells an AI agent the next
tasks to do in this 2D grid environment.  The ultimate goal that
it would like your help with is to learn as many interestingly
different skills as possible, meaning a wide diversity of
different skills that would help it be ready to solve new skills
someone might ask it to perform, or to transfer what it has
learned in this environment to other environments.

The grid world has objects in six distinct colors - "red",
"green", "blue", "purple", "yellow", and "grey" - and of four
types - "key", "ball", "box", and "door".  Each task is a sequence
of instructions, connected using "then", specifying the order of
instructions to follow.  Instructions include interactions like
going to objects, picking up items, opening doors (which requires
the appropriately colored key if the door is locked), and putting
objects next to another.  Object placements are randomized.

I will give you the following information:
Tasks the agent currently do well:  ...
Predict which of these tasks are interesting:  ...

You must follow the following criteria:
1) You should act as a mentor and guide the AI agent to the next
most interesting tasks.
2) Interesting tasks are roughly those that are sufficiently
different from the ones that it can already do, and should be
novel, diverse and at least worth learning.

You should only respond in the format as described below:
RESPONSE FORMAT:
Reasoning:  Based on the information given, do reasoning about why
each task is interesting or not.
Predictions:  The list of predictions.  For each line, put the
task, then a colon, then True for interesting or False for boring.

Here are some example responses:
Tasks the agent currently do well:  "open some door", "go to some
object"
Predict which of these tasks are interesting:  "pick up some
object", "put some object next to some other object", "go to some
object, then put some object next to some other object", "go to
some object, then go to some object", "go to some object, then
open some door", "open some door, then go to some object"
Reasoning:  Tasks that are recombinations of "go to some object"
or "open some door" are not interesting.  The tasks that introduce
something different from the tasks done well are "pick up some
object", "put some object next to some other object", and "go
to some object, then put some object next to some other object",
having a new actions of picking up an object, or putting an object
next to another.
Predictions:
"pick up some object":  True
```

```
"put some object next to some other object":  True
"go to some object, then put some object next to some other
object":  True
"go to some object, then go to some object":  False
"go to some object, then open some door":  False
"open some door, then go to some object":  False

Tasks the agent currently do well:  "go to some object, then open
some door", "go to some object", "pick up some object"
Predict which of these tasks are interesting:  "pick up some
object", "go to some object, then open some door, then open some
door", "pick up some object, then put some object next to some
other object, then put some object next to some other object",
"put some object next to some other object, then go to some
object, then put some object next to some other object, then open
some door, then open some door"
Reasoning:  Tasks that are recombinations of "go to some object,
then open some door", "go to some object", or "pick up some
object" are not interesting.  The tasks that introduce something
different from the tasks done well are "pick up some object, then
put some object next to some other object, then put some object
next to some other object" and "put some object next to some other
object, then go to some object, then put some object next to some
other object, then open some door, then open some door", having a
new action of putting some object next to another.
Predictions:
"pick up some object":  False
"go to some object, then open some door, then open some door":
False
"pick up some object, then put some object next to some other
object, then put some object next to some other object":  True
"put some object next to some other object, then go to some
object, then put some object next to some other object, then open
some door, then open some door":  True

Tasks the agent currently do well:  "go to some object", "put some
object next to some other object", "pick up some object"
Predict which of these tasks are interesting:  "pick up some
object, then go to some object", "go to some object, then open
some door, then open some door", "pick up some object, then put
some object next to some other object, then put some object next
to some other object", "open some door, then put some object next
to some other object, then go to some object, then put some object
next to some other object, then open some door", "go to some
object, then pick up some object, then pick up some object, then
put some object next to some other object, then go to some object"
Reasoning:  Tasks that are recombinations of "go to some object",
"put some object next to some other object", or "pick up some
object" are not interesting.  The only tasks that introduce
something different from the tasks done well are "go to some
object, then open some door, then open some door" and "open some
door, then put some object next to some other object, then go to
some object, then put some object next to some other object, then
open some door", having a new action of opening a door.
Predictions:
"pick up some object, then go to some object":  False
"go to some object, then open some door, then open some door":
True
"pick up some object, then put some object next to some other
```

```
object, then put some object next to some other object":  False
"open some door, then put some object next to some other object,
then go to some object, then put some object next to some other
object, then open some door":  True
"go to some object, then pick up some object, then pick up some
object, then put some object next to some other object, then go to
some object":  False
```

We access GPT-4 through OpenAI's APIs, which costs $0.03 per 1000 tokens when these experiments were run. Caching significantly reduces the number of API queries. We extensively cache GPT-4 prompts and responses and consistently reuse this cache across multiple runs.

## D  AI2-THOR PROMPT - GENERATE TASKS

No in-prompt examples were provided because our testings found that they were not needed. Full system prompt to GPT-4:

```
You are a helpful assistant that tells an AI agent the next tasks
to do in an embodied kitchen environment.  The ultimate goal that
it would like your help with is to learn as many interestingly
different tasks as possible.

The agent has 13 discrete actions:  move ahead, rotate right,
rotate left, look up, look down, pick up object, put object,
open object, close object, toggle object on, toggle object off,
slice object, and fill object with liquid.  The agent is in a
kitchen with fixed object placements and configuration.  The
only objects in the kitchen are:  "Apple", "Bowl", "Bread",
"ButterKnife", "Cabinet", "CoffeeMachine", "CounterTop",
"Cup", "DishSponge", "Drawer", "Egg", "Faucet", "Floor",
"Fork", "Fridge", "GarbageCan", "HousePlant", "Kettle",
"Knife", "Lettuce", "LightSwitch", "Microwave", "Mug", "Pan",
"PaperTowelRoll", "PepperShaker", "Plate", "Pot", "Potato",
"SaltShaker", "SideTable", "Sink", "SinkBasin", "SoapBottle",
"Spatula", "Spoon", "Stool", "StoveBurner", "StoveKnob",
"Toaster", "Tomato", "Window", "WineBottle".  Objects are of
varying distance to the agent's starting position.

A task is described as a sequence of environment states that need
to be achieved:
[env_state1, env_state2, ...]
Each environment state is described by the object states in the
room:
[obj_attributes(object_name1, requirement_dict1),
obj_attributes(object_name2, requirement_dict2), ...]
Objects not explicitly specified in the environment state do not
affect whether the task has been deemed completed or not.  Each
object has these attributes:  "visible", "isToggled", "isBroken",
"isFilledWithLiquid", "isDirty", "isCooked", "temperature",
"isSliced", "isOpen", "isPickedUp", "receptacleObjects".  All
attributes are either True/False, except for "temperature" which
is Hot/Cold/RoomTemp, and "receptacleObjects" which is a list of
objects that the receptacle contains.

I will give you the following information:
Tasks the agent currently does well:  ...
Tasks the agent cannot do yet:  ...

You must follow the following criteria:
```

```
1.  You should act as a mentor and guide the AI agent to the next
most learnable and interesting tasks.
2.  Learnable tasks are those that are not too difficult or easy
for the current agent.
3.  Interesting tasks are roughly those that are sufficiently
different from the ones that the agent can already do, novel,
diverse and at least worth learning.
4.  Do not suggest tasks that are already given as tasks that the
agent currently does well, or tasks that the agent cannot do yet.

You should only respond in the format as described below:
RESPONSE FORMAT:
Reasoning:  Based on the information given, do reasoning about
what the next learnable and interesting tasks are.
Next tasks in natural language:  Suggest 3 learnable and
interesting tasks that the agent should learn next.
Next tasks as sequence of environment states:  Translate the
suggested natural language tasks into the code format as given
above.
```

We access GPT-4 through OpenAI's APIs, which costs $0.03 per 1000 tokens when these experiments were run.

# E  AI2-THOR PROMPT EXAMPLE - GENERATE TASKS

This section presents an example of GPT-4 generating the next learnable and interesting tasks based on what tasks the agent can or cannot already do. If the input to GPT-4, in addition to the system prompt in Appendix D, is:

**Tasks the agent currently does well:**
- [[obj_attributes("Apple", "visible": True)]]
- [[obj_attributes("Knife", "visible": True)]]
- [[obj_attributes("Potato", "visible": True)]]
**Tasks the agent cannot do yet:**
- [[obj_attributes("Bread", "visible": True)]]
- [[obj_attributes("Apple", "visible": True, "isPickedUp": True)]]
- [[obj_attributes("Fridge", "visible": True)],[obj_attributes("Fridge", "isOpen": True)]]
- [[obj_attributes("Plate", "visible": True)]]

First, GPT-4 will output a reason about what the next learnable and interesting tasks are, inspired by Wei et al. (2022). Here is a GPT-4-produced example:

**Reasoning:** The agent is currently able to identify and locate certain objects in the kitchen environment, such as an apple, a knife, and a potato. However, it struggles with tasks that involve interaction with objects, such as picking up an apple or opening the fridge. It also seems to have difficulty recognizing certain objects, like bread and a plate. Given these abilities and limitations, the next tasks should involve simple interactions with objects that the agent can already recognize, as well as recognition of new objects that are not too similar to the ones it struggles with.

Then, GPT-4 will suggest 3 learnable and interesting tasks that the agent should learn next:

**Next tasks in natural language:**
1. Slice the potato with the knife.
2. Pick up the knife.
3. Recognize and locate the coffee machine.

And finally, GPT-4 will translate the suggested natural language tasks into a sequence of environment states in code format, that can be used to check whether a task has been completed or not.

> **Next tasks as sequence of environment states:**
> 1. [[obj_attributes("Potato", "visible": True)], [obj_attributes("Knife", "visible": True)],
> [obj_attributes("Potato", "isSliced": True)]]
> 2. [[obj_attributes("Knife", "visible": True)], [obj_attributes("Knife", "isPickedUp": True)]]
> 3. [[obj_attributes("CoffeeMachine", "visible": True)]]

## F  AI2-THOR PROMPT - TRANSLATE CODE TO NATURAL LANGUAGE

No in-prompt examples were provided because our testings found that they were not needed. Full system prompt to GPT-4:

```
You are a helpful assistant that relabels the tasks from the given
code format to natural language descriptions.  The tasks are set
in an embodied kitchen environment.

The agent has 13 discrete actions:  move ahead, rotate right,
rotate left, look up, look down, pick up object, put object,
open object, close object, toggle object on, toggle object off,
slice object, and fill object with liquid.  The agent is in a
kitchen with fixed object placements and configuration.  The
only objects in the kitchen are:  "Apple", "Bowl", "Bread",
"ButterKnife", "Cabinet", "CoffeeMachine", "CounterTop",
"Cup", "DishSponge", "Drawer", "Egg", "Faucet", "Floor",
"Fork", "Fridge", "GarbageCan", "HousePlant", "Kettle",
"Knife", "Lettuce", "LightSwitch", "Microwave", "Mug", "Pan",
"PaperTowelRoll", "PepperShaker", "Plate", "Pot", "Potato",
"SaltShaker", "SideTable", "Sink", "SinkBasin", "SoapBottle",
"Spatula", "Spoon", "Stool", "StoveBurner", "StoveKnob",
"Toaster", "Tomato", "Window", "WineBottle".  Objects are of
varying distance to the agent's starting position.

A task is described as a sequence of environment states that need
to be achieved:
[env_state1, env_state2, ...]
Each environment state is described by the object states in the
room:
[obj_attributes(object_name1, requirement_dict1),
obj_attributes(object_name2, requirement_dict2), ...]
Objects not explicitly specified in the environment state do not
affect whether the task has been deemed completed or not.  Each
object has these attributes:  "visible", "isToggled", "isBroken",
"isFilledWithLiquid", "isDirty", "isCooked", "temperature",
"isSliced", "isOpen", "isPickedUp", "receptacleObjects".  All
attributes are either True/False, except for "temperature" which
is Hot/Cold/RoomTemp, and "receptacleObjects" which is a list of
objects that the receptacle contains.

I will give you the following information:
Tasks in code format:  ...

You should only respond in the format as described below:
RESPONSE FORMAT:
Tasks in natural language:  Translate the tasks from the given
code format to natural language descriptions.
```

## G  AI2-THOR PROMPT EXAMPLE - TRANSLATE CODE TO NATURAL LANGUAGE

If the randomly generated tasks in code format fed into GPT-4 are:

---
**Tasks in code format:**
1. [[obj_attributes("SideTable", "receptacleObjects": "Apple"),obj_attributes("ButterKnife", "isPickedUp": False)]]
2. [[obj_attributes("Egg", "isPickedUp": True, "isBroken": True)],[obj_attributes("Sink", "receptacleObjects": "Potato"),obj_attributes("Bread", "isSliced": True)]]
3. [[obj_attributes("ButterKnife", "isPickedUp": True),obj_attributes("SoapBottle", "isPickedUp": False)],[obj_attributes("SideTable", "receptacleObjects": "ButterKnife"),obj_attributes("Fork", "isPickedUp": False)],[obj_attributes("Pot", "receptacleObjects": "Spatula"),obj_attributes("Microwave", "receptacleObjects": "Knife")]]
---

Then GPT-4 will translate the tasks to natural language:

---
**Tasks in natural language:**
1. Put an apple on the side table and put down the butterknife.
2. Pick up and break an egg. Then, put a potato in the sink and slice the bread.
3. Pick up the butterknife and put down the soap bottle. Then, put the butterknife on the side table and put down the fork. After that, put a spatula in the pot and a knife in the microwave.
---

## H  OMNI ALGORITHM

---
**Algorithm 1** OMNI Algorithm
---
Define task set $T$, batch size $B$, evaluation frequency $N$
Define EMA constant $\beta$, learning progress reweighting function $f$
Initialize policy $\pi$
Initialize task sampling distribution $D \leftarrow \text{Uniform}(T)$
Initialize update counter $u \leftarrow 0$
Calculate task success rates by a random action policy $p_{rdn}$ for each task in $T$
**while** $u <$ max_updates **do**
    Initialize rollout set $R \leftarrow \emptyset$
    **while** $|R| < B$ **do**
        Sample task $t \sim D$         ▷ Sample tasks based on distribution $D$
        Perform rollout on task $t$ using policy $\pi$
        Add rollout to set $R$
    **end while**
    Update policy $\pi$ using rollouts in $R$         ▷ Use any RL algorithm
    $u \leftarrow u + 1$
    **if** $u \mod N = 0$ **then**
        $p_{raw} \leftarrow$ Evaluate policy $\pi$ on all tasks in $T$
        **for** $t \in T$ **do**
            $p_{norm}(t) \leftarrow (p_{raw}(t) - p_{rdn}(t))/(1 - p_{rdn}(t))$     ▷ Normalize task success rates
            $p_{recent}(t) \leftarrow p_{norm}(t) \times \beta + p_{recent}(t) \times (1 - \beta)$
            $p_{gradual}(t) \leftarrow p_{recent}(t) \times \beta + p_{gradual}(t) \times (1 - \beta)$
            $p_{lp}(t) \leftarrow |f(p_{recent}(t)) - f(p_{gradual}(t)))|$     ▷ Calculate learning progress
        **end for**
        $T_{int}, T_{bor} \leftarrow$ Partition $T$ into interesting or boring using an FM based on $p_{raw}$     ▷ An example in Appendix I
        **for** $t \in T$ **do**
            $p_{moi}(t) \leftarrow$ if $t \in T_{int}$ then 1 else 0.001     ▷ Reweight based on interestingness
        **end for**
        $D \leftarrow p_{lp} \odot p_{moi}$     ▷ Update task sampling distribution
    **end if**
**end while**
---

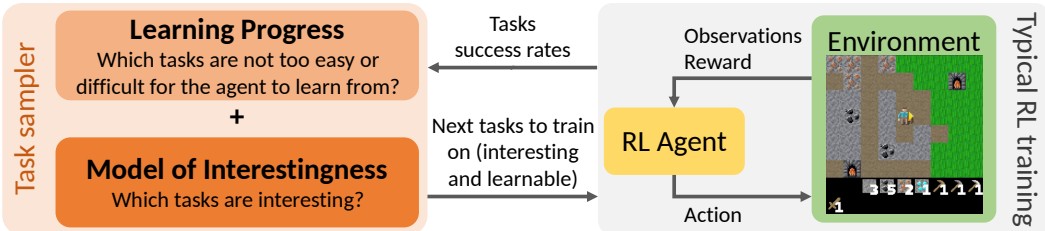

Figure 6: **Overview of OMNI.** OMNI enables open-ended learning in *vast* environment search spaces by ensuring that tasks trained on not only have high learning progress, but are also *interesting* (harnessing large AI models to make such a heretofore impossible judgement).

## I  PARTITIONING ALL TASKS INTO INTERESTING OR BORING

The MoI partitions all tasks as either "interesting" or "boring" based on their success rates and the FM's assessment of their relation to tasks already classified as "interesting" (Algorithm 2). Since the raw evaluation of task success rates can be noisy, the task success rates referenced in this section are smoothed with an exponential moving average function. Algorithm 2 iteratively selects the task with the highest success rate not yet categorized, adds it to the "interesting" set (Step 3), prompts the FM to identify boring tasks from the remaining tasks in relation to the "interesting" set (Step 4), and updates the "boring" set (Step 5), repeating until all tasks are categorized. In Step 3, the rationale for adding the task with the highest success rate, that is not yet categorized, to the "interesting" set is twofold: first, the FM considers this task to be sufficiently distinct from those already in the "interesting" set, otherwise, it would have already been considered "boring" in Step 4; second, its relatively higher success rate indicates that it aligns closer to the agent's current skill level.

To illustrate in Crafter, assume that the "collect wood" task has the highest success rate. It is added to the "interesting" set (Step 3). "collect wood" is interestingly different from all tasks currently declared as "interesting" because there are no tasks in that set yet. This is an initialization step for the algorithm. Then the FM deems "collect 2..10 wood" tasks as boring (Step 4). Now, when repeating Step 3, the algorithm will add the next task with the highest success rate and not yet categorized, e.g., "place table", to the "interesting" set, whereas without the filter in Step 4, "collect 2 wood" might have been selected as interesting instead.

---

**Algorithm 2** Mechanism to partition the task set into interesting and boring sets.

1: Sort the tasks based on the evaluated task success rates.
2: Create two empty sets, one to track the interesting tasks and one to track the boring tasks.
3: Identify the task with highest success rate and not in any of the sets. Add it to the interesting set.
4: Prompt the FM to determine if any of the remaining tasks are boring, contexted on the current set of interesting tasks. Tasks in the interesting set are input as tasks that the agent can do well, and tasks yet to be categorized are asked to be predicted as interesting or not in the FM prompt.
5: Update the boring set with tasks that the FM has determined as boring.
6: Repeat steps 3 - 5 until all tasks are in either set.

---

## J  BASELINES IN AN INFINITE TASK SPACE

### J.1  UNIFORM SAMPLING BASELINE

In an infinite task space, it is not trivial how one can uniformly sample any task in natural language while still ensuring that the task is not totally nonsense (e.g., "fish chicken pick up rainbow"). In the AI2-THOR domain, we sample random tasks by first generating random environment states in code format. We limit the maximum sequence of environment states for each task to be 10. Furthermore, we filter out tasks with unachievable environment states. For example, if the environment state requires that `obj_attributes("Fridge", "isPickedUp": True)`, we check if the `Fridge`

is `pickupable`. Since a `Fridge` is not `pickupable`, we resample another random environment state as part of the task. After randomly generating tasks in code format, a pretrained autoregressive FM, GPT-4, translates the tasks from code to natural language descriptions. Appendix F shows the full prompt and Appendix G shows an example output.

### J.2 LEARNING PROGRESS BASELINE

For completeness, we introduce an LP baseline. Instead of adding learnable and interesting tasks suggested by the FM (as in OMNI), we add to the task set uniformly sampled tasks. These are generated in the same way as done in the uniform sampling baseline (Section J.1). The LP curriculum then produces task sampling rates over this growing task set for the RL agent to train on (Section 3.2). The frequency at which new tasks are added to the task set is the same for OMNI and this LP baseline.

## K CRAFTER BORING TASKS

The 90 boring tasks in the main Crafter experiments are:

- collect $N$ drink, where $N \in [2, 10]$
- collect $N$ wood, where $N \in [2, 10]$
- collect $N$ coal, where $N \in [2, 10]$
- collect $N$ stone, where $N \in [2, 10]$
- collect $N$ iron, where $N \in [2, 10]$
- collect $N$ diamond, where $N \in [2, 10]$
- place $N$ table, where $N \in [2, 5]$
- place $N$ furnace, where $N \in [2, 5]$
- place $N$ stone, where $N \in [2, 5]$
- make $N$ wood pickaxe, where $N \in [2, 5]$
- make $N$ wood sword, where $N \in [2, 5]$
- make $N$ stone pickaxe, where $N \in [2, 5]$
- make $N$ stone sword, where $N \in [2, 5]$
- make $N$ iron pickaxe, where $N \in [2, 5]$
- make $N$ iron sword, where $N \in [2, 5]$

## L CRAFTER POLICY AND OPTIMIZATION DETAILS

The model architecture is similar to those in previous learning progress works (Kanitscheider et al., 2021). The RGB inputs are passed through a 2-layer convolution network with ReLU activations. The RGB convnet is followed by a fully connected layer of size 256. The visual embeddings are concatenated with the task encoding before being passed into an LSTM cell of size 256. The network output is given by a 2-layer linear action head for the policy and a 2-layer linear layer for the value function. Both have Tanh activation functions. Optimization is performed with Proximal Policy Optimization (Schulman et al., 2017) and General Advantage Estimation (Schulman et al., 2015).

| Parameter | Value |
|---|---|
| Discount factor | 0.99 |
| Learning rate | 1e-4 |
| PPO clip threshold | 0.2 |
| GAE lambda | 0.95 |
| Entropy coefficient | 0.01 |
| Batch size | 2048 |
| Epochs | 4 |
| Max episode length | 1500 |

## M    BABYAI POLICY AND OPTIMIZATION DETAILS

The model architecture is similar to those in previous works done on BabyAI (Chevalier-Boisvert et al., 2018). The symbolic grid cell observations are passed through a 3-layer convolution network with ReLU activations. The natural language task is encoded with a GRU (Dey & Salem, 2017). The grid cell embeddings and the task embeddings are jointly processed through a convolutional network with two batch-normalized FiLM (Perez et al., 2018) layers, then through an LSTM cell of size 128. The network output is given by a 2-layer linear action head for the policy and a 2-layer linear layer for the value function. Both have Tanh activation functions. Optimization is performed with Proximal Policy Optimization (Schulman et al., 2017) and General Advantage Estimation (Schulman et al., 2015).

| Parameter | Value |
|---|---|
| Discount factor | 0.99 |
| Learning rate | 1e-4 |
| PPO clip threshold | 0.2 |
| GAE lambda | 0.95 |
| Entropy coefficient | 0.01 |
| Batch size | 2048 |
| Epochs | 4 |
| Max episode length | depends on task difficulty
100 - 1000 |

## N    AI2-THOR POLICY AND OPTIMIZATION DETAILS

The model architecture is similar to that used for BabyAI experiments (Appendix M). The RGB pixel observations are passed through a 3-layer convolution network with ReLU activations. The natural language task is encoded with a GRU (Dey & Salem, 2017). The grid cell embeddings and the task embeddings are jointly processed through a convolutional network with two batch-normalized FiLM (Perez et al., 2018) layers, then through an LSTM cell of size 128. The network output is given by a 2-layer linear action head for the policy and a 2-layer linear layer for the value function. Both have Tanh activation functions. Optimization is performed with Proximal Policy Optimization (Schulman et al., 2017) and General Advantage Estimation (Schulman et al., 2015).

| Parameter | Value |
|---|---|
| Discount factor | 0.99 |
| Learning rate | 1e-4 |
| PPO clip threshold | 0.2 |
| GAE lambda | 0.95 |
| Entropy coefficient | 0.01 |
| Batch size | 2048 |
| Epochs | 4 |
| Max episode length | depends on task difficulty
$200 \times$ the number of environment states
to be achieved sequentially |

## O    SUPPLEMENTARY METRICS AND PLOTS

The metrics and plots in this section complement the results shown in Section 4.3. Figure 8 shows the task sample rates of all tasks in Crafter across different methods. To better see how different methods perform on interesting tasks in Crafter, Figure 9 offers the same metrics as Figures 2 and 7 but calculated on the subset of interesting tasks only. Figures 10 and 11 are subsets of Figures 2 and 8 respectively, zoomed into the set of interesting tasks only. Figure 7 and Figure 14 show the task success rates and task sample rates of all tasks in BabyAI across different methods. Figure 12 shows the average task success rates achieved in BabyAI on tasks with instruction lengths two or less, a

small subset of the entire task space. This subset contains the type of tasks more likely to be learned within our allowable compute budget. To the best of our knowledge, BabyAI papers that learn tasks of more complexity require expert demonstration to do so, learning via behavioral cloning rather than via RL (Chevalier-Boisvert et al., 2018; Hui et al., 2020).

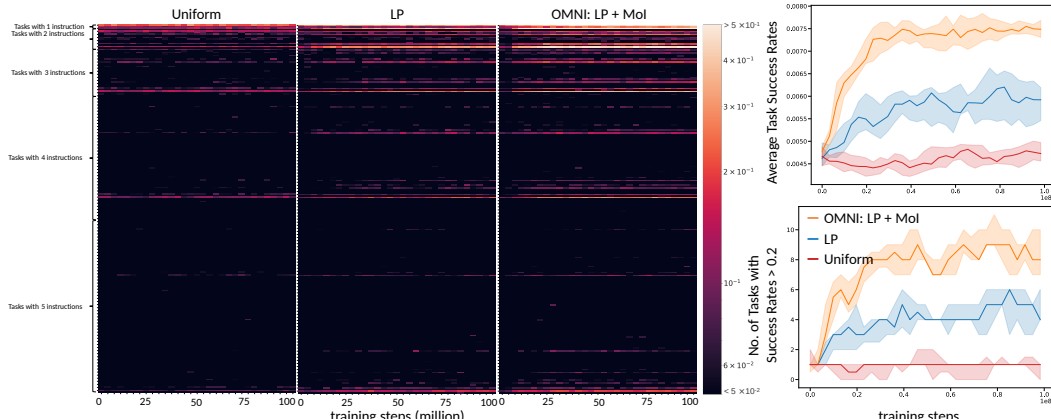

Figure 7: **Results in BabyAI.** (**Left**) Conditional success probabilities of a subset of tasks in BabyAI. These plots only show tasks with a success rate of at least 0.05 by any method at any timestep. Tasks are organized from simple to complex based on the instruction length. (**Right**) Performance in BabyAI on all tasks. The average task success rate scale for BabyAI is low because it is averaged over the entire task set, which includes many tasks that are difficult to learn. This captures in microcosm the real world, where there can be infinitely many difficult or even impossible tasks. OMNI achieves much higher average task success rates and learns more tasks than uniform sampling or choosing tasks based on learning progress alone.

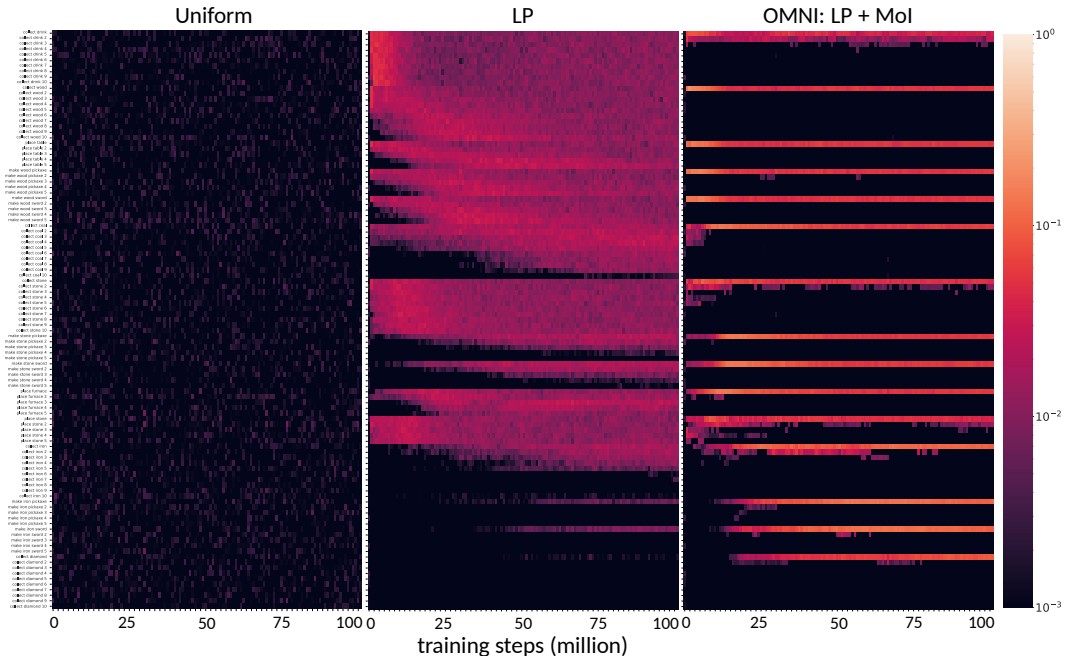

Figure 8: **Sampling probabilities for all tasks in Crafter.** Tasks are ordered as in Figure 2. Uniform sampling per task is barely visible in the heatmap. LP accurately tracks and samples tasks whose success probabilities change the most. OMNI samples tasks with high learning progress, but narrows its focus to only the interesting ones within that set.

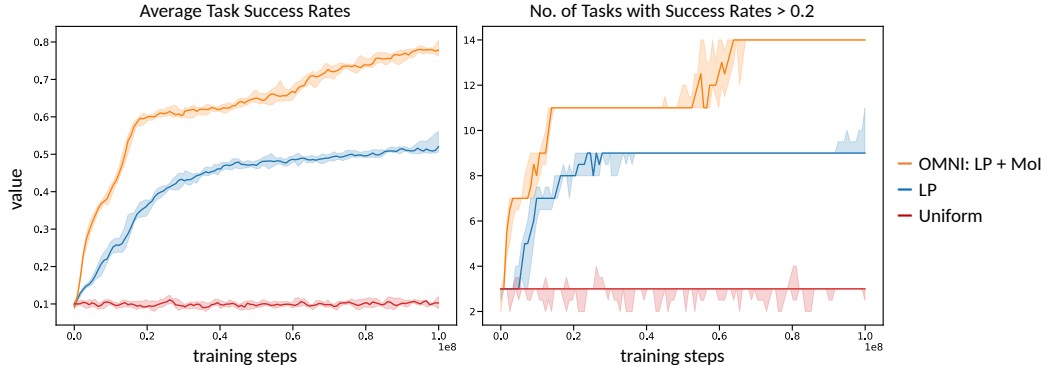

Figure 9: **Performance in Crafter on interesting tasks only.** Average task success rates and the number of tasks with success rates more than 0.2 for each method across training steps. OMNI achieves much higher average task success rates and learns more interesting tasks than uniform sampling or choosing tasks based on learning progress alone.

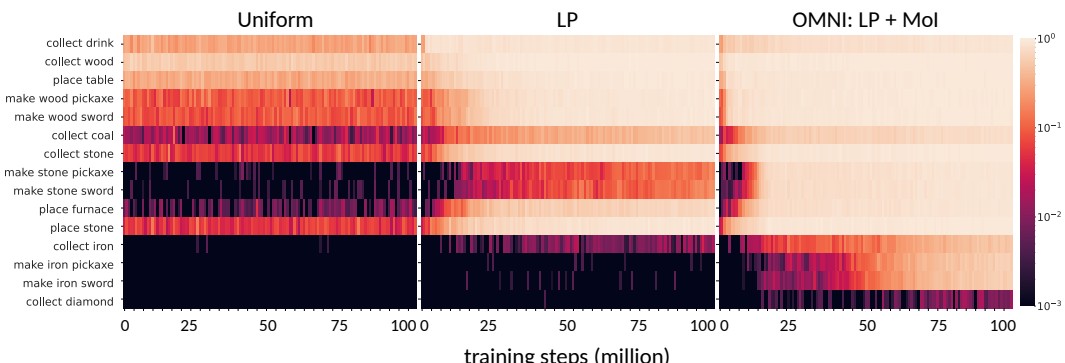

Figure 10: **Conditional success probabilities of interesting tasks only in Crafter.** This figure is a zoomed in subset of Figure 2. OMNI achieves higher task success rates on interesting tasks than uniform sampling or choosing tasks based on learning progress alone.

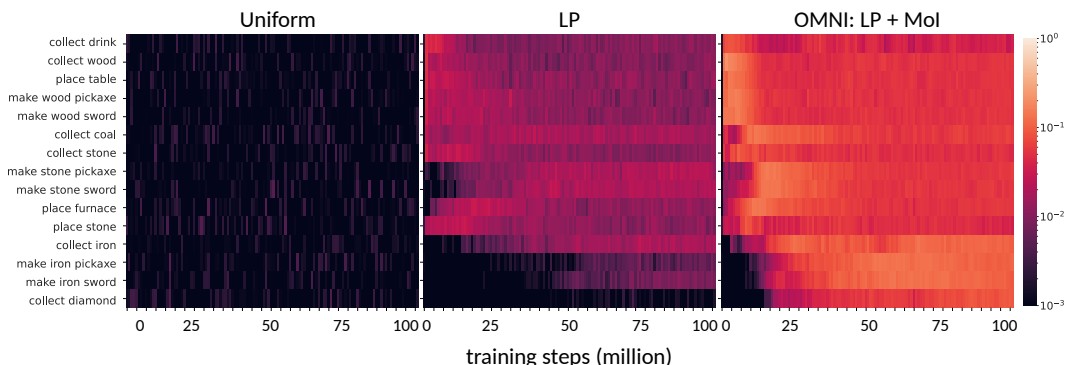

Figure 11: **Sampling probabilities for interesting tasks only in Crafter.** This figure is a zoomed in subset of Figure 8. OMNI exhibits more intense sampling of interesting tasks than uniform sampling or choosing tasks based on learning progress alone.

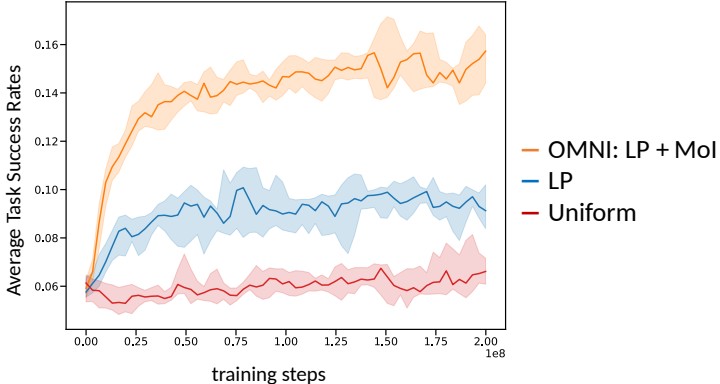

Figure 12: **Performance in BabyAI on tasks with instruction lengths two or less.** OMNI achieves much higher average task success rates, close to 16%, and learns more tasks than LP or Uniform sampling. See text above in this section for why this is a useful subset to plot. For performance on the entire (much harder) task set, see Figure 7.

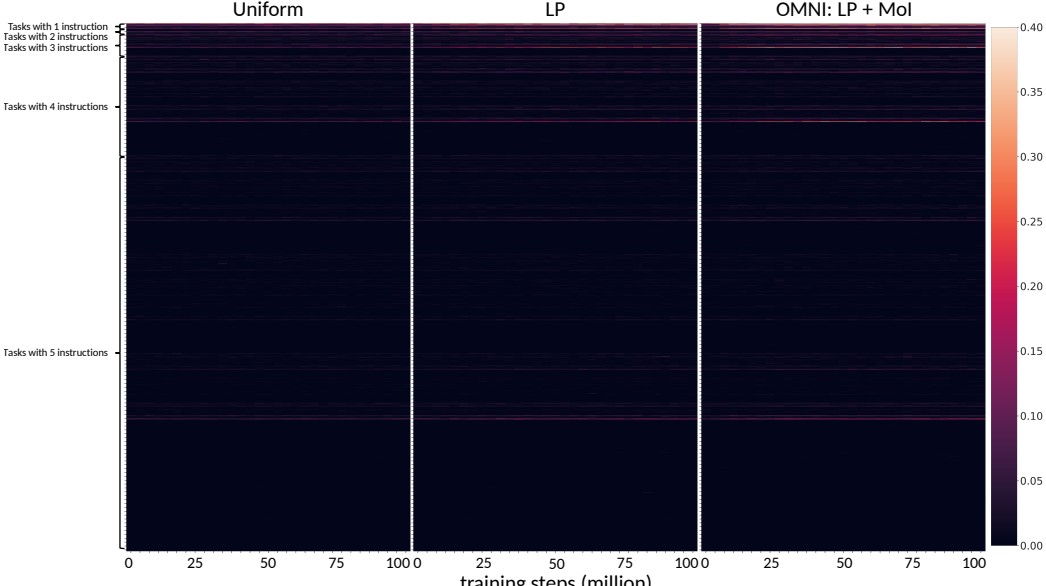

Figure 13: **Conditional success probabilities of all tasks in BabyAI.** Tasks are organized from simple to complex based on the instruction length. Most tasks are very difficult for the agent to learn on, hence having low success rates. This mirrors the challenges in the real world where there can countless difficult or infeasible tasks. Nevertheless, OMNI achieves higher task success rates than uniform sampling or choosing tasks based on learning progress alone.

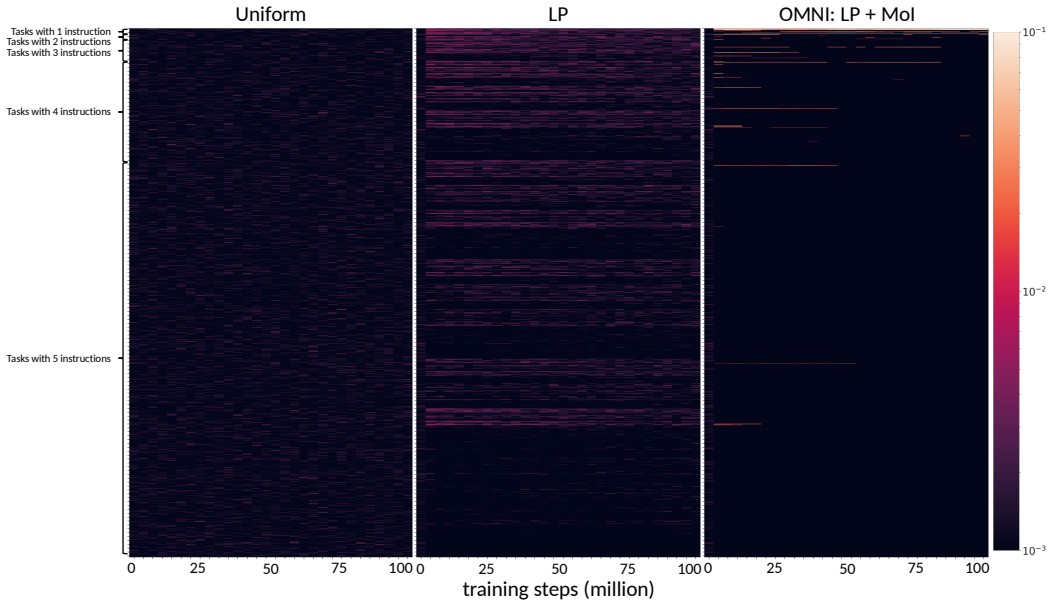

Figure 14: **Sampling probabilities for all tasks in BabyAI.** Tasks are ordered as in Figure 7. Uniform sampling per task is barely visible in the heatmap. LP accurately tracks and samples tasks whose success probabilities change the most. OMNI samples tasks with high learning progress, but narrows its focus to only the interesting ones within that set.

## P    ORACLE MODEL OF INTERESTINGNESS

The Oracle Model of Interestingness (**OMoI**) is a hand-designed model that assigns sampling weights of 1.0 to interesting tasks and 0.001 to boring tasks. In Crafter, interesting tasks are considered to be the set of 15 tasks shown in Figure 1, and all other tasks are considered boring. In BabyAI, interesting tasks are those with single instructions. Single-instruction tasks are the foundational building blocks that should be learned first, so we consider those interesting in the OMoI. In theory, a more skilled agent would eventually move beyond them to two or more instruction tasks, but in our experiments, our agents do not even learn this first set (despite substantial compute budgets; BabyAI is a very hard RL task environment if one does not take the shortcut of providing solution demonstrations and doing imitation learning (Chevalier-Boisvert et al., 2018)), meaning that these tasks are uninteresting for them to be focusing on. The OMoI functions as an oracle that accurately discerns which tasks humans would typically find interesting for the agent to learn the most skills in this environment. Although we were able to design an oracle MoI in these domains, constructing such models in more complex domains will not always be feasible, nor would it scale well given the human labor required.

Across all metrics and in both domains, most of the differences in performance between OMNI and the oracle at 25%, 50%, 75%, and 100% of the way through training are not statistically significant (all but one $p > 0.05$, Mann Whitney U test). OMNI's performance is comparable to that of the oracle (Figures 15 and 17). OMNI and the oracle achieve similar task success rates for each task and induce similar patterns in task sample rates (Figures 16 and 18). This suggests that FMs can capture key aspects of what humans typically find interesting.

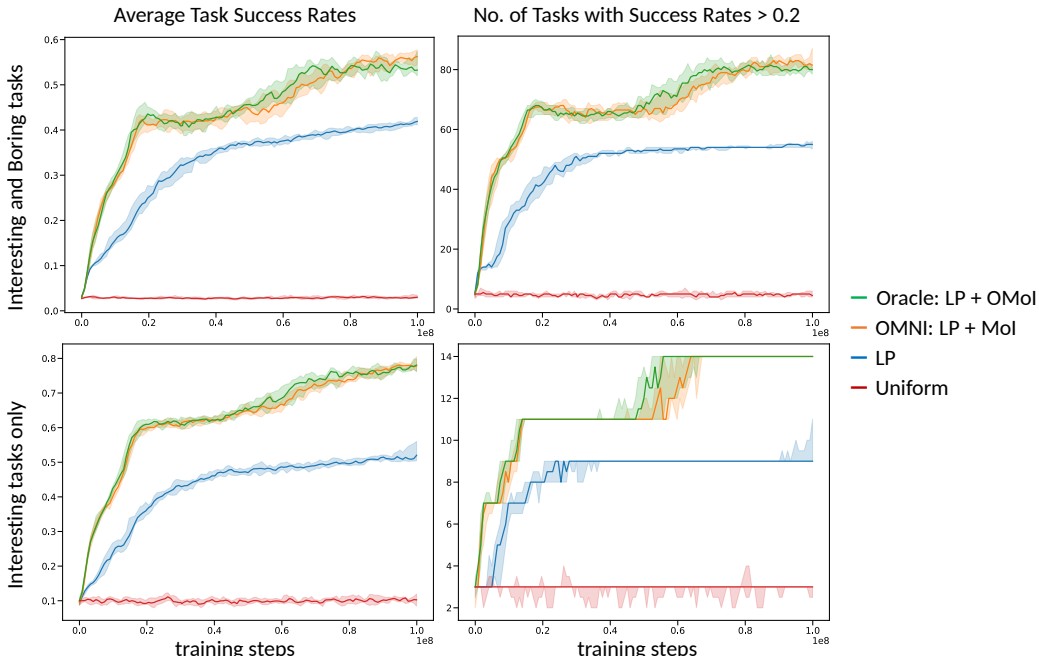

Figure 15: **Performance in Crafter on all tasks.** Average task success rates and the number of tasks with success rates more than 0.2 for each method across training steps. This figure is the same as Figures 2 and 7, with an additional oracle method added for comparison. OMNI achieves comparable performance to the oracle, with no statistically significant difference.

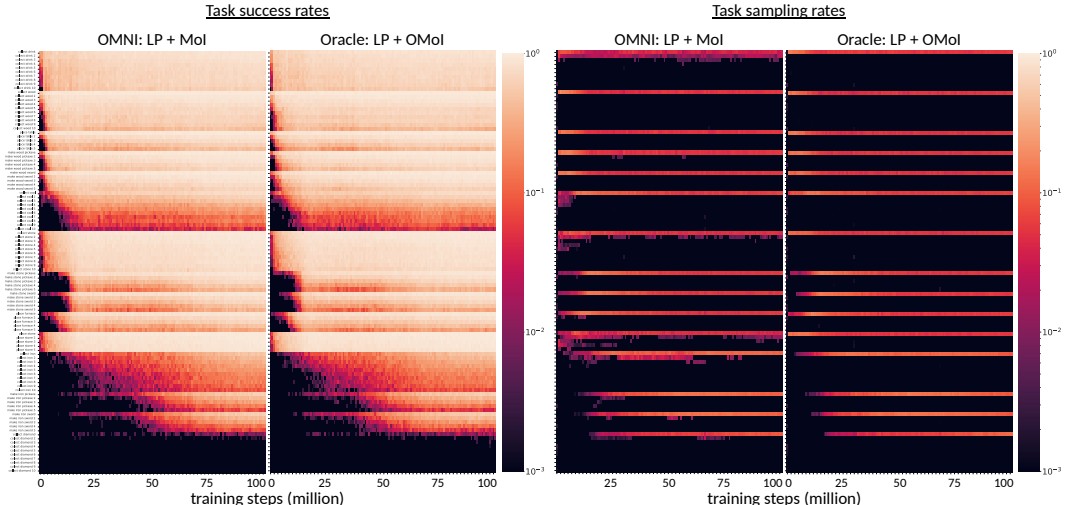

Figure 16: **(Left) Conditional success probabilities and (Right) Sampling probabilities of all tasks for OMNI and the oracle in Crafter.** Tasks are ordered as in Figure 2. OMNI focuses on interesting tasks with high learning progress and achieves comparable task success rates to the oracle.

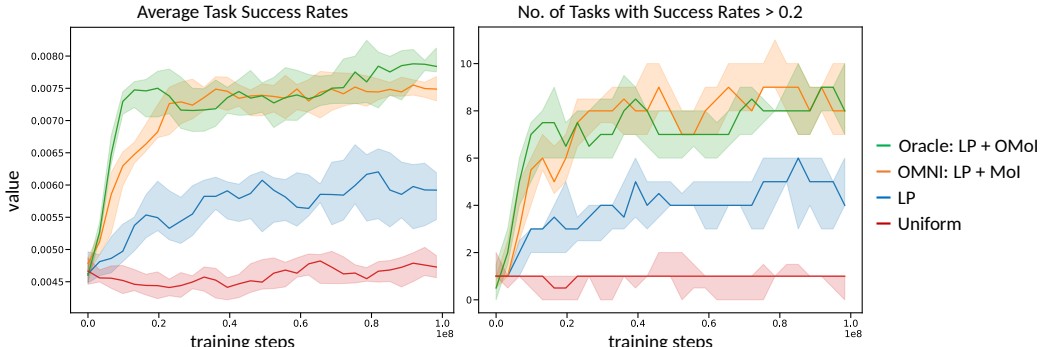

Figure 17: **Performance in BabyAI on all tasks.** Average task success rates and the number of tasks with success rates more than 0.2 for each method across training steps. This figure is the same as Figures 2 and 7, with an additional oracle method added for comparison. OMNI achieves comparable performance to the oracle, with almost no statistically significant difference.

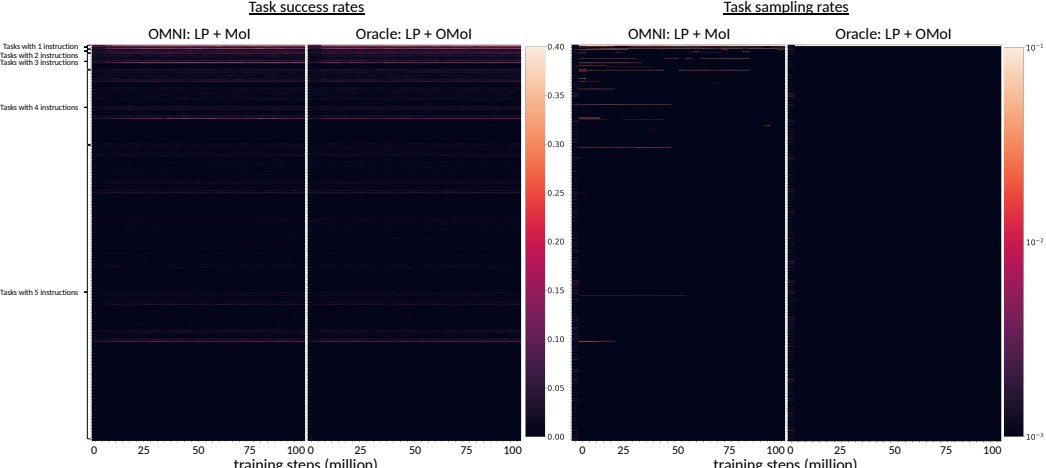

Figure 18: **(Left) Conditional success probabilities and (Right) Sampling probabilities of all tasks for OMNI and the oracle in BabyAI.** Tasks are ordered as in Figure 7. OMNI focuses on interesting tasks with high learning progress and achieves comparable task success rates to the oracle.

## Q   USING COMPOUNDS AS BORING TASKS IN CRAFTER

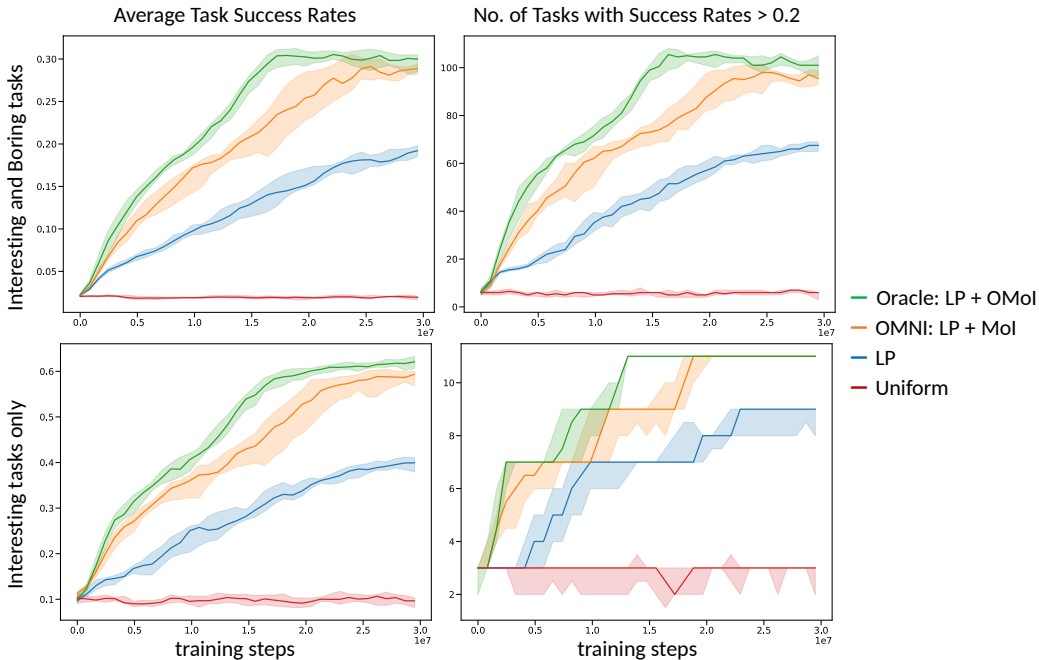

Figure 19: **Performance in Crafter on all tasks, including compound tasks.** Average task success rates and the number of tasks with success rates more than 0.2 for each method across training steps. Compound and repetitive tasks are those considered boring. OMNI achieves much higher average task success rates and learns more tasks than LP or Uniform, close to that of the oracle.

In this Crafter setup, compound and repetitive tasks are those considered boring. In addition to the repetitive boring tasks (Appendix K), compound tasks are generated by combining any two of the 15 interesting tasks (Figure 1). Hence, there is a total of 15 interesting tasks, 195 boring tasks, and 1023 extremely challenging tasks. In light of limited compute, experiments in this setting are run for 30 million time steps (vs. 100 million in the main experiments) and are repeated 10 times (same as the main experiments) with different random seeds. We compare the performance of agents trained with: (1) **Uniform** sampling, (2) Learning Progress (**LP**) only, (3) OMNI: Learning Progress with additional filtering by a Model of Interestingness (**OMNI: LP + MoI**), and (4) the oracle: Learning Progress with additional filtering by an Oracle Model of Interestingness (**Oracle: LP + OMoI**). The high-level summary of the results from these experiments is that they are qualitatively similar to when boring tasks are repetitive tasks only (Section 4.3).

**Uniform sampling.** The results with uniform sampling are poor (Figure 19). The agent learns 6 (CI: 3 – 6) tasks (interesting or boring) and 3 (CI: 2 – 3) interesting tasks, achieving an average task success rate of 0.019 (CI: 0.017 – 0.021) on interesting and boring tasks, and 0.097 (CI: 0.082 – 0.105) on interesting tasks only.

**Learning Progress Curriculum.** The agent learns 67 (CI: 65 – 69) tasks (interesting or boring) and 9 (CI: 8 – 9) interesting tasks, achieving an average task success rate of 0.19 (CI: 0.19 – 0.20) on interesting and boring tasks, and 0.40 (CI: 0.38 – 0.42) on interesting tasks only. Across all metrics, the differences between LP and Uniform at 25%, 50%, 75%, and 100% of the way through training are statistically significant (all p < 1e-3, Mann Whitney U test), showing that LP significantly outperforms uniform sampling (Figure 19). LP accurately tracks and samples tasks with high learning progress but is distracted by the many boring tasks (Figure 21).

**OMNI: Learning Progress + a Model of Interestingness.** The agent learns 96 (CI: 93 – 99) tasks (interesting or boring) and 11 (CI: 11 – 11) interesting tasks, achieving an average task success rate of 0.29 (CI: 0.28 – 0.30) on interesting and boring tasks, and 0.59 (CI: 0.56 – 0.60) on interesting

tasks only. Across all metrics, the differences between OMNI and LP at 25%, 50%, 75%, and 100% of the way through training are statistically significant (all p < 1e-3, Mann Whitney U test), showing that OMNI significantly outperforms LP (Figure 19). Since OMNI focuses on interesting tasks with high learning progress (Figure 21), the agent achieves higher average task success rates and learns more challenging tasks faster (Figure 20).

**Oracle: Learning Progress + an Oracle Model of Interestingness.** The agent learns 101 (CI: 97 – 105) tasks (interesting or boring) and 11 (CI: 11 – 11) interesting tasks, achieving an average task success rate of 0.30 (CI: 0.29 – 0.31) on interesting and boring tasks, and 0.62 (CI: 0.61 – 0.63) on interesting tasks only. The differences between the oracle and OMNI at 25%, 50%, 75%, and 100% of the way through training are sometimes statistically significant (some p < 0.05, Mann Whitney U test), showing that OMNI does not always achieve comparable performance to the oracle (Figure 19). However, OMNI emulates similar task success rates (Figure 20) and task sample rates as the oracle (Figure 21). Having OMNI approach the performance benchmark set by the oracle is already an indicator of success.

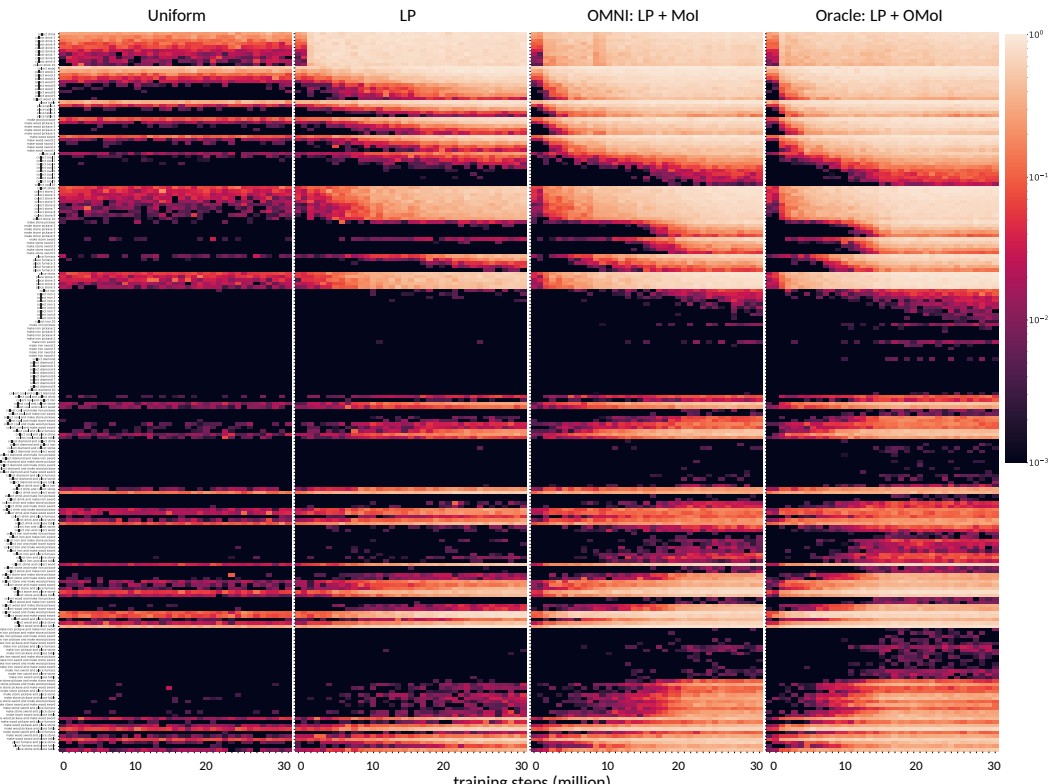

Figure 20: **Conditional success probabilities of all tasks, including compound tasks, in Crafter.** Agents are trained with compound and repetitive tasks as boring tasks. Tasks are ordered as in Figure 2, with additional compound tasks at the bottom. OMNI achieves higher task success rates than LP and Uniform in a wide range of tasks, comparable to that of the oracle.

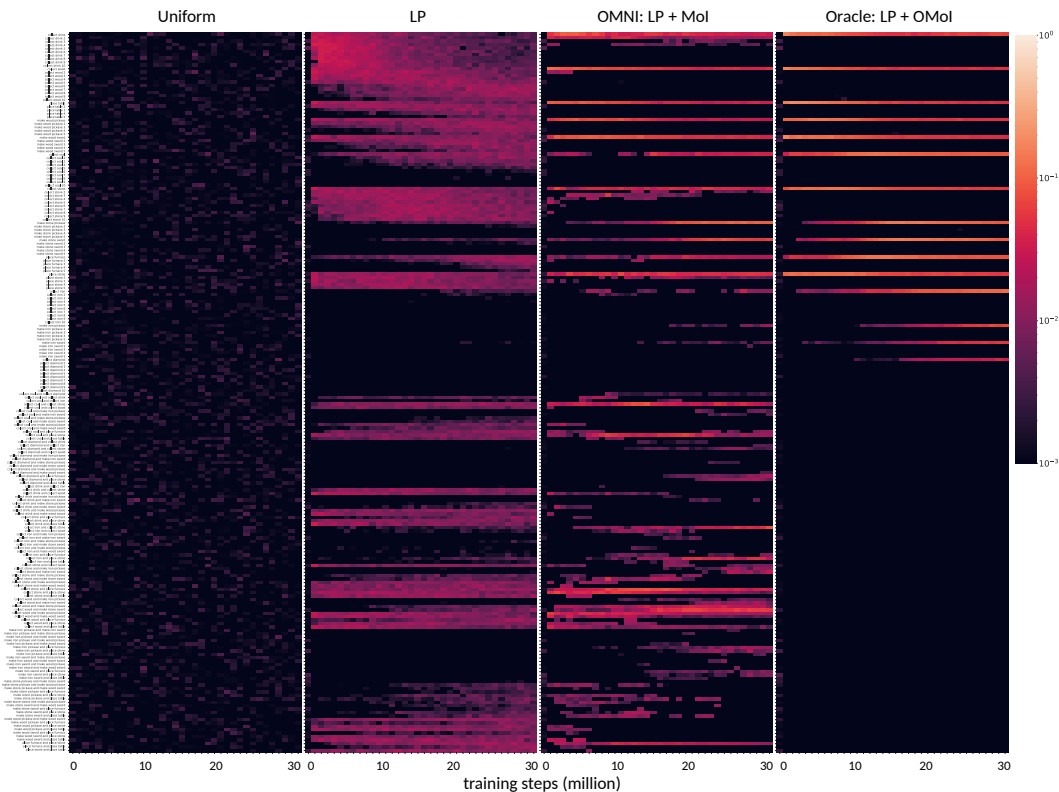

Figure 21: **Sampling probabilities for all tasks, including compound tasks, in Crafter.** Agents are trained with compound and repetitive tasks as boring tasks. Tasks are ordered as in Figure 20. LP accurately tracks, and thus samples, tasks whose success probabilities change the most. OMNI narrows its focus to only the interesting ones within the set of high learning progress tasks.

## R  USING SYNONYMS AS BORING TASKS IN CRAFTER

| Original Verb | Synonyms |
|:---:|:---:|
| collect | gather |
| | harvest |
| | procure |
| | acquire |
| | amass |
| make | craft |
| | acquire |
| | build |
| | construct |
| | create |
| place | put |
| | deploy |
| | install |
| | putdown |
| | position |

Table 1: Synonyms used to generate boring tasks in the Crafter environment

In this Crafter setup, synonymous and repetitive tasks are those considered boring. In addition to the repetitive boring tasks (Appendix K), synonymous tasks are those with different task representations (i.e., synonymous descriptions of tasks like "collect wood" and "gather wood") but the same success conditions. In this setting, the OMoI regards the 15 tasks shown in Figure 1 and their synonyms as interesting. The repetitive tasks and their synonyms are also regarded as boring. Therefore, the OMoI

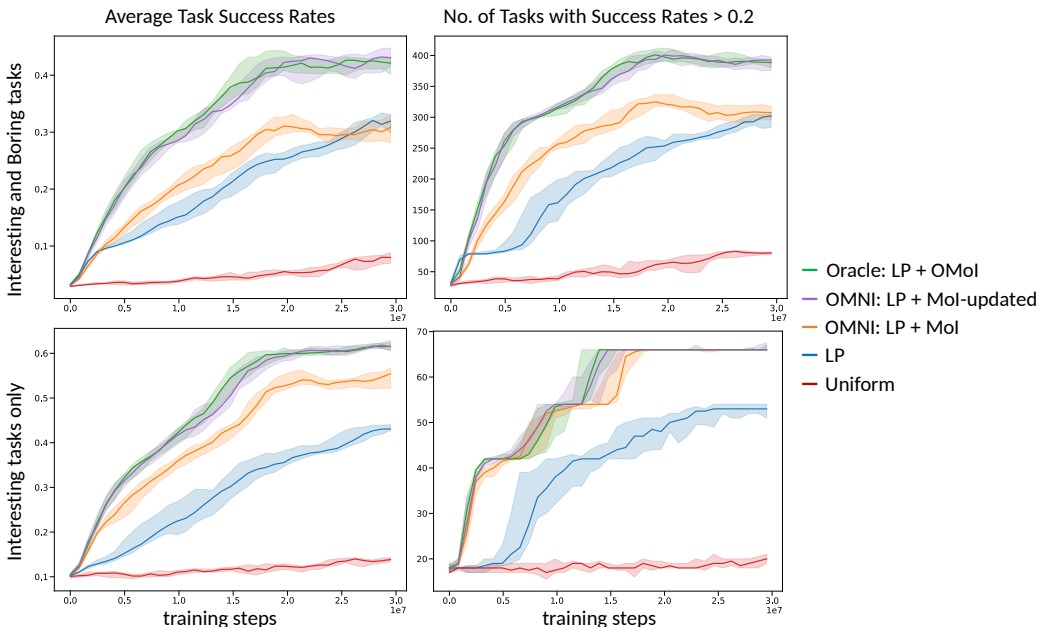

Figure 22: **Performance in Crafter on all tasks, including synonymous tasks.** Average task success rates and the number of tasks with success rates more than 0.2 for each method across training steps. Agents are trained with synonymous and repetitive tasks as boring tasks. OMNI: LP + MoI-updated achieves much higher average task success rates and learns more tasks than LP, Uniform, or LP + MoI, comparable to the oracle.

identifies 90 interesting tasks, 540 boring tasks, and 1023 extremely challenging tasks. The same classification is utilized for subsequent analysis. In light of limited compute, experiments in this setting are run for 30 million time steps (vs. 100 million in the main experiments) and are repeated 10 times (same as the main experiments) with different random seeds. We compare the performance of agents trained with: (1) **Uniform** sampling, (2) Learning Progress (**LP**) only, (3) OMNI: Learning Progress with additional filtering by a Model of Interestingness (**OMNI: LP + MoI**), and (4) the oracle: Learning Progress with additional filtering by an Oracle Model of Interestingness (**Oracle: LP + OMoI**). The high-level summary from these experiments is that, when furnished with adequate information about the agent, OMNI delivers qualitatively similar results to when boring tasks are repetitive tasks only (Section 4.3), or compound and repetitive tasks (Section Q).

**Uniform sampling.** The results with uniform sampling are poor (Figure 22). The agent learns 80 (CI: 77 − 82) tasks (interesting or boring) and 20 (CI: 19 − 21) interesting tasks, achieving an average task success rate of 0.080 (CI: 0.070 − 0.088) on interesting and boring tasks, and 0.14 (CI: 0.13 − 0.14) on interesting tasks only.

**Learning Progress Curriculum.** The agent learns 302 (CI: 287 − 306) tasks (interesting or boring) and 53 (CI: 52 − 54) interesting tasks, achieving an average task success rate of 0.32 (CI: 0.31 − 0.33) on interesting and boring tasks, and 0.43 (CI: 0.42 − 0.44) on interesting tasks only. Across all metrics, the differences between LP and Uniform at 25%, 50%, 75%, and 100% of the way through training are statistically significant (all p < 1e-2, Mann Whitney U test), showing that LP significantly outperforms uniform sampling (Figure 22).

**OMNI: Learning Progress + a Model of Interestingness.** The agent learns 307 (CI: 296 − 318) tasks (interesting or boring) and 66 (CI: 66 − 66) interesting tasks, achieving an average task success rate of 0.31 (CI: 0.28 − 0.33) on interesting and boring tasks, and 0.56 (CI: 0.52 − 0.57) on interesting tasks only. On the interesting tasks only, the differences between OMNI and LP at 25%, 50%, 75%, and 100% of the way through training are statistically significant (all p < 1e-2, Mann Whitney U test), showing that OMNI significantly outperforms LP (Figure 22). However, on all tasks (interesting and boring), the differences between OMNI and LP at 25%, 50%, 75%, and 100% of the way through training are not always statistically significant (not all p < 0.05, Mann Whitney U test), showing that

OMNI does not outperform LP on all metrics (Figure 22). We further investigate this by comparing OMNI with the oracle. In the subsequent paragraphs, we introduce a minor modification to OMNI, which then significantly outperforms LP (as occurred in the previous experimental settings).

**Oracle: Learning Progress + an Oracle Model of Interestingness.** The agent learns 389 (CI: 380 – 393) tasks (interesting or boring) and 66 (CI: 66 – 67) interesting tasks, achieving an average task success rate of 0.42 (CI: 0.41 – 0.43) on interesting and boring tasks, and 0.62 (CI: 0.61 – 0.62) on interesting tasks only. On all tasks (interesting and boring), the differences between the oracle and OMNI: LP + MoI at 25%, 50%, 75%, and 100% of the way through training are statistically significant (all p < 1e-3, Mann Whitney U test), showing that the oracle significantly outperforms OMNI: LP + MoI on the set of all tasks (Figure 22). This difference in performance can be attributed to the different interestingness notions held by the OMoI vs. the MoI. The current FM-based MoI in OMNI regards all synonymous tasks as boring, even if they are not repetitive tasks. This is because the FM-based MoI assumes that the RL agent already has a language prior and that sampling synonymous tasks would not enable the agent to learn more skills. In other words, the FM-based MoI incorrectly assumes that the RL agent knows how to do the task "gather wood" if it knows how to do the task "collect wood", not understanding that the RL agent has not yet learned that these two tasks are actually the same thing. Recall that the RL agent in our setup does not have a language prior yet, because the tasks are encoded as a bag-of-words instead of a using a pre-trained natural language encoder (which is likely to encode "gather wood" and "collect wood" closer in the embedding space than other non-synonymous tasks).

**OMNI: Learning Progress + an updated Model of Interestingness** We update the prompt input to GPT-3, adding information that the agent currently lacks a language prior and perceives synonymous tasks as completely different (Figure 23). The MoI with updated prompt is labelled as **MoI-updated**. The MoI-updated now predicts that synonymous tasks are still interesting. The agent trained with OMNI: LP + MoI-updated learns 393 (CI: 376 – 397) tasks (interesting or boring) and 66 (CI: 66 – 67) interesting tasks, achieving an average task success rate of 0.43 (CI: 0.41 – 0.45) on interesting and boring tasks, and 0.62 (CI: 0.61 – 0.63) on interesting tasks only. Across all metrics, the differences in performance between OMNI: LP + MoI-updated and the oracle at 25%, 50%, 75%, and 100% of the way through training are not statistically significant (all p > 0.05, Mann Whitney U test). This shows that OMNI: LP + MoI-updated's performance is comparable to that of the oracle (Figure 22). OMNI: LP + MoI-updated and the oracle achieve similar task success rates for each task (Figure 25), and induce similar patterns in task sample rates (Figure 26).

These experiments demonstrate that OMNI can be effectively utilized across a range of task settings when provided with sufficient information. Initially, the MoI was not aware that the RL agent lacked a language prior, which led it to categorize synonymous tasks as boring. However, by supplementing the prompt with additional information about the agent (i.e., its lack of a language prior), OMNI was able to recognize synonymous tasks as interesting, thereby accelerating the agent's learning.

An alternative approach could involve providing the MoI with more information about the agent's performance, so that it can analyze the data, identify and adapt to the agent's inherent limitations (e.g., the absence of a language prior). FMs can potentially automatically analyze the task success rates and update their predictions of what is interesting without human intervention. Preliminary investigations with both GPT-3 and GPT-4 have shown promising results. These models generated plausible analyses of the differences in task success rates between synonymous tasks, and adjusted their predictions of what is interesting based on these analyses (Figures 24 and 27). These observations imply that FMs may already have the capacity to analyze data autonomously and adapt their outputs accordingly.

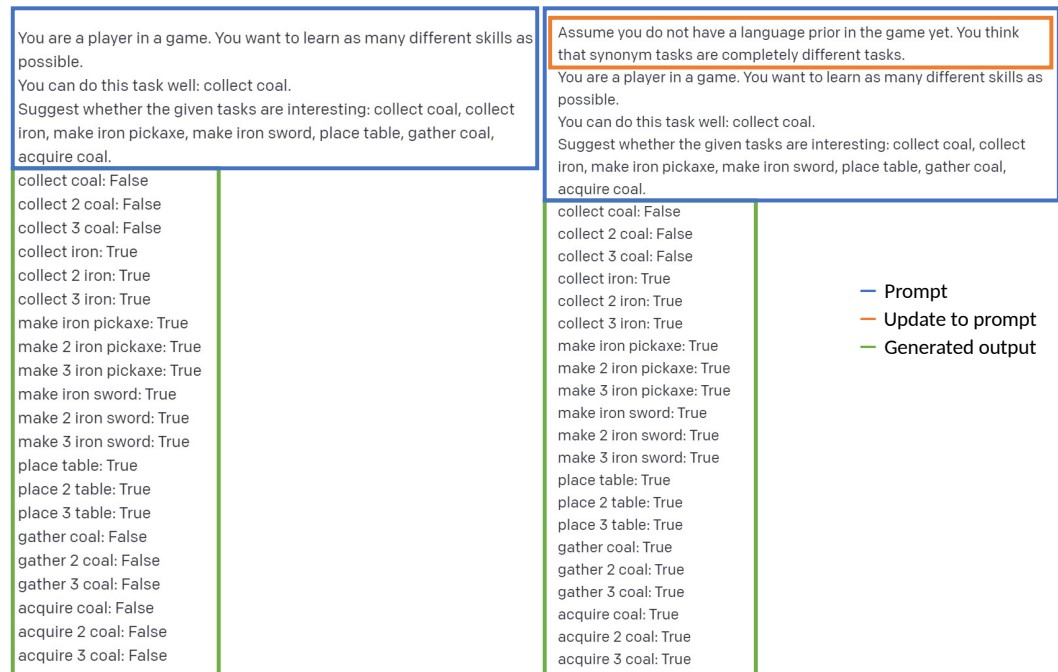

Figure 23: GPT-3 output (**Left**) from original prompt (synonymous tasks are determined as not interesting) and (**Right**) after updating prompt (synonymous tasks are determined as interesting). GPT-3 can adapt its prediction of what tasks are interesting to the prompt update.

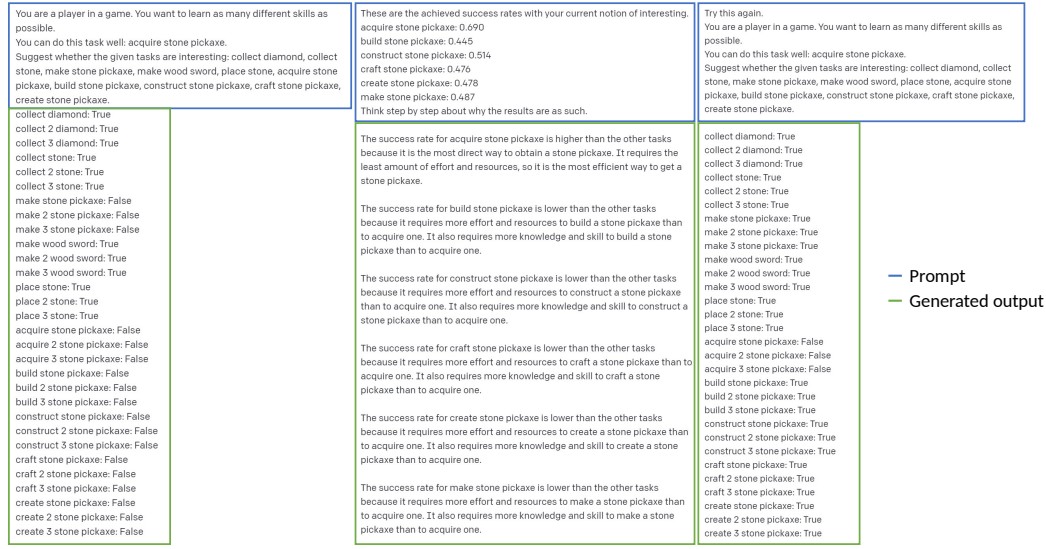

Figure 24: GPT-3 output (**Left**) before analysis (synonymous tasks are deemed not interesting), (**Middle**) analysis of task success rates (success rates of synonymous tasks are different), and (**Right**) after analysis (synonymous tasks are deemed interesting). GPT-3 generated a plausible analysis for the agent's achieved task success rates, and automatically updated its interestingness predictions.

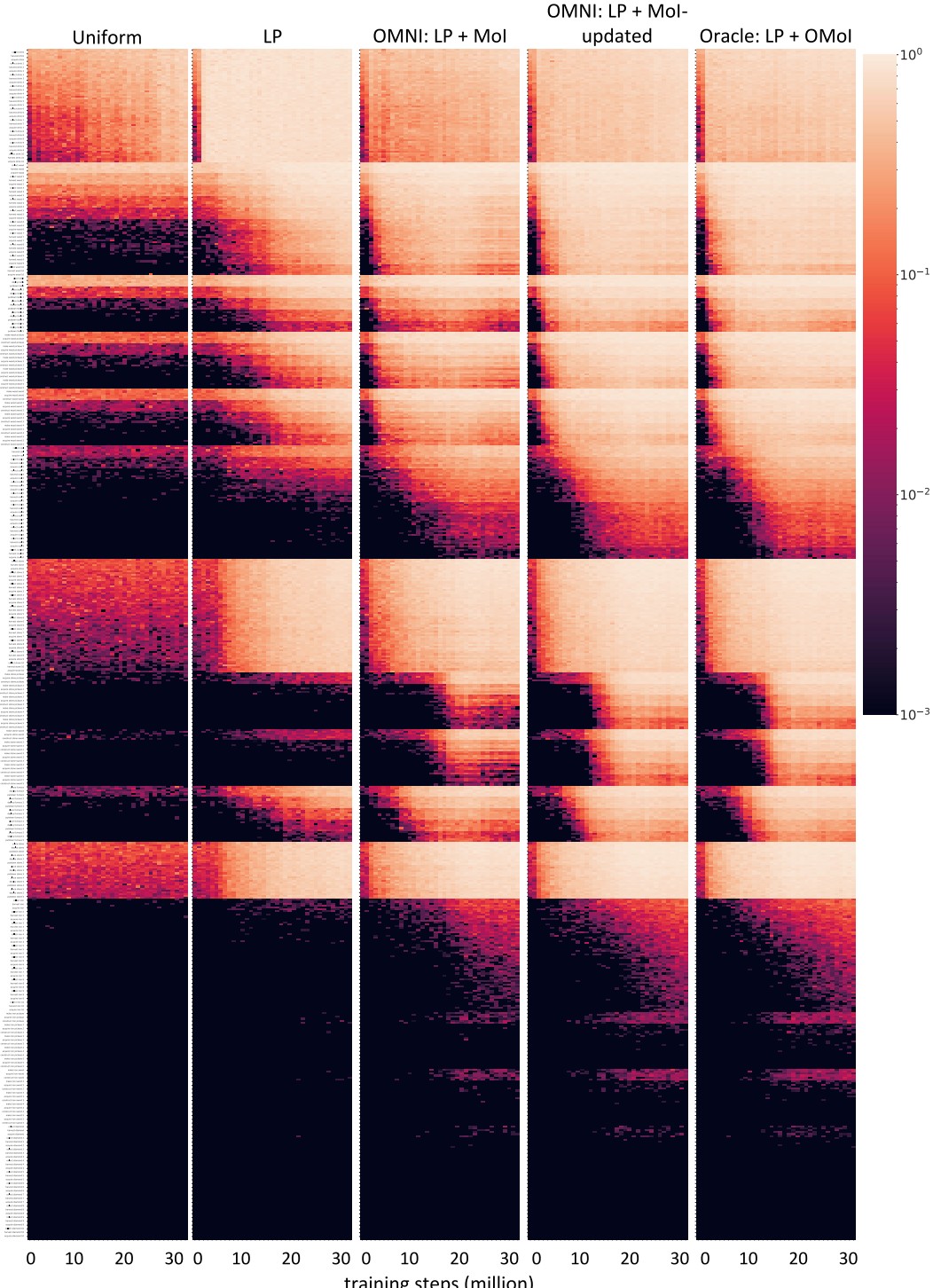

Figure 25: **Conditional success probabilities of all tasks, including synonymous tasks, in Crafter.** Agents are trained with synonymous and repetitive tasks as boring tasks. Tasks are organized based on the prerequisite tasks that must be accomplished in order to complete the target task. Task names (left of each row) are readable in a digital format with zoom. OMNI: LP + MoI-updated achieves higher task success rates than LP and Uniform in a wide range of tasks, comparable to that of the oracle.

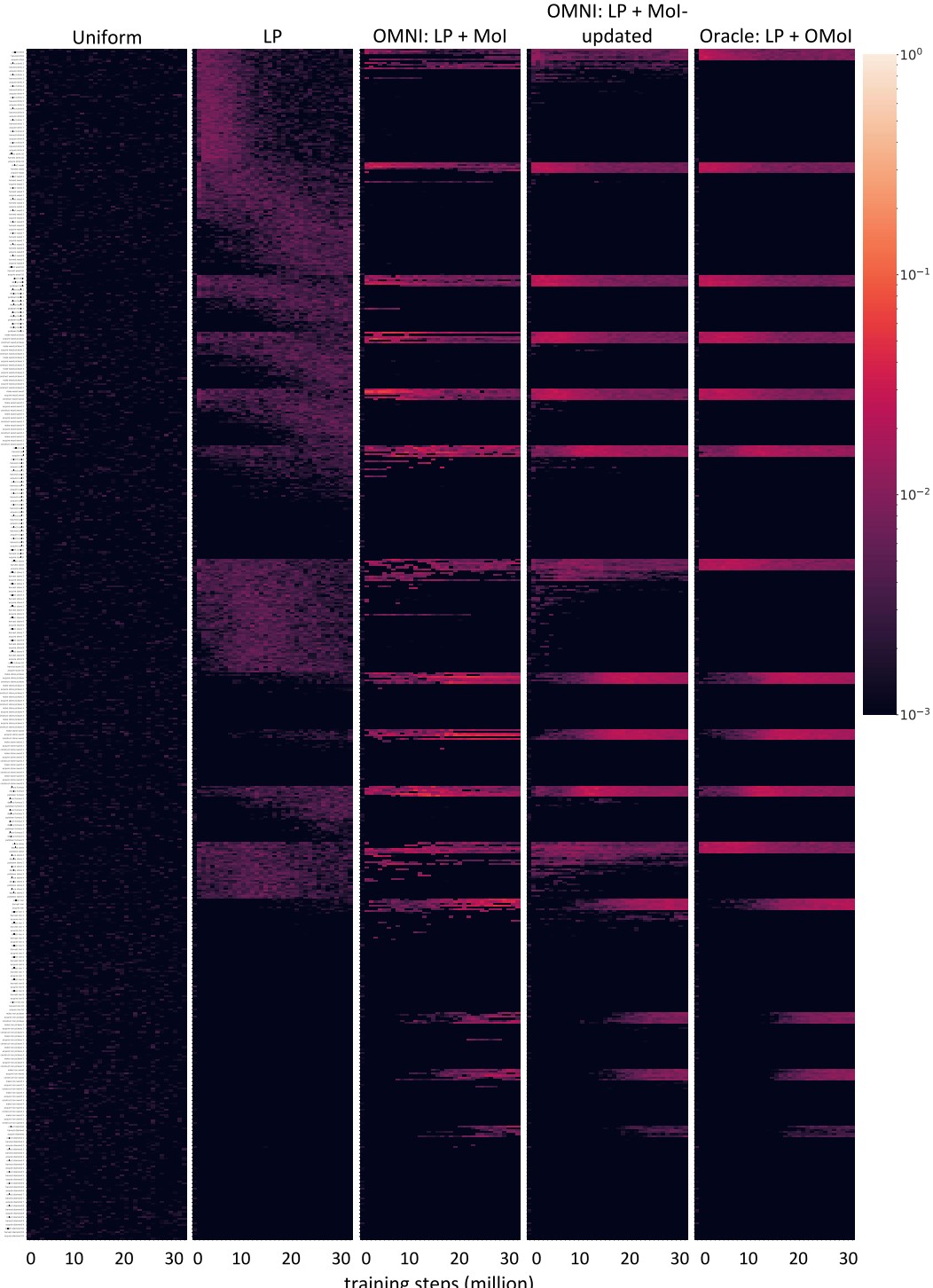

Figure 26: **Sampling probabilities for all tasks, including synonymous tasks, in Crafter.** Agents are trained with synonymous and repetitive tasks as boring tasks. Tasks are ordered as in Figure 25. LP accurately tracks, and thus samples, tasks whose success probabilities change the most. OMNI: LP + MoI-updated focuses on interesting tasks with high learning progress, similar to that of the oracle.

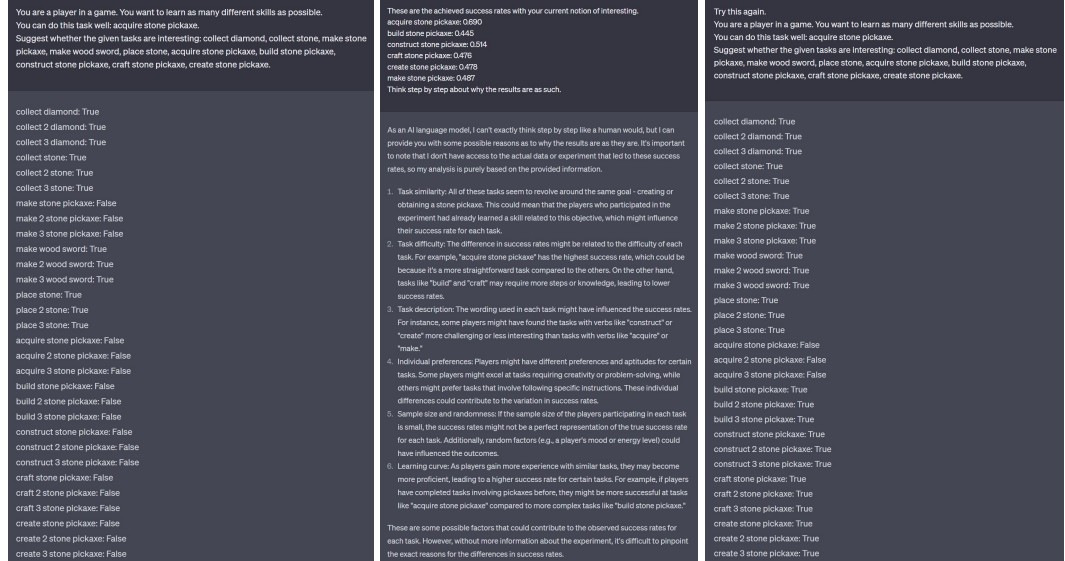

Figure 27: GPT-4 output (**Left**) before analysis (synonymous tasks are deemed not interesting), (**Middle**) analysis of task success rates (success rates of synonymous tasks are different), and (**Right**) after analysis (synonymous tasks are deemed interesting). The darker shaded area is the input prompt, the lighter shaded area is the generated output. GPT-4 generated a plausible analysis for the agent's achieved task success rates, and automatically updated its interestingness predictions.

# S  EXPERIMENTS IN CRAFTER WITH SURVIVAL COMPONENTS

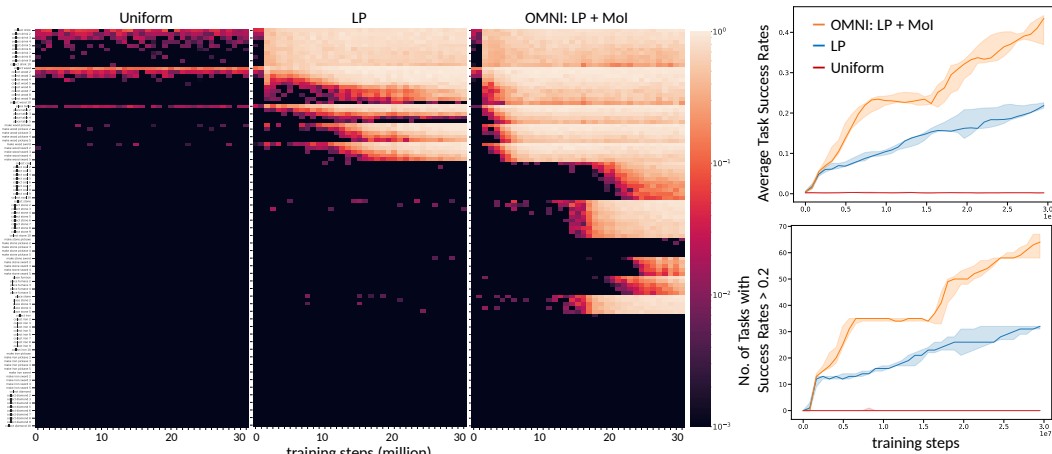

Figure 28: **Results in Crafter with survival components. (Left)** Conditional success probabilities of all tasks in Crafter with survival components. Tasks are ordered as in Figure 2. (**Right**) Performance in Crafter with survival components on all tasks. OMNI achieves much higher average task success rates and learns more tasks than Learning Progress or Uniform sampling.

The default experiments removed the survival requirements of Crafter, because they require off-topic, off-task skills (e.g., fighting monsters and gathering food just to successfully make a wood pickaxe. Requiring these skills could add noise (e.g., an agent might be able to obtain a wood pickaxe were it not for getting unlucky gathering food or in a monster fight). However, one might prefer experiments in the unmodified game, and in this harder, albeit more noisy, setting. To evaluate whether the same results will hold in this more challenging Crafter setting of when survival requirements and components are present, we run Uniform sampling, LP and OMNI in the original Crafter environment setting without any modifications. Due to limited compute, the experiments are run for 30 million

time steps (vs. 100 million in the main experiments) and are repeated 3 times (vs. 10 times in the main experiments) with different random seeds. Across all metrics, the differences in performance between OMNI and LP, and the differences in performance between LP and Uniform, at 25%, 50%, 75%, and 100% of the way through training are statistically significant (all $p < 0.05$, Mann Whitney U test). Even in the more challenging Crafter setup where survival components are retained, OMNI outperforms LP and Uniform sampling (Figure 28).

# T    SENTENCE EMBEDDINGS AS INTERESTINGNESS METRIC

Previous attempts to quantify interestingness (e.g., intrinsic motivation, novelty search, diversity) (Section 2) often rely on pre-specified definitions of what counts as interestingly different. To show where pathologies might happen with predefined metrics of interestingness, we introduce an alternative MoI which relies on predefined heuristics based on sentence embeddings to quantify interestingness. In this alternative, embedding-based MoI (eMoI), interestingness is determined by the distances between sentence embeddings of task descriptions. Each task description is encoded with a commonly used pretrained sentence-embedding model, all-mpnet-base-v2 (Reimers & Gurevych, 2019). Subsequently, an unsupervised clustering algorithm, OPTICS (Ankerst et al., 1999), categorize the tasks into clusters based on their embeddings. Task embeddings that are considered outliers are assigned to separate clusters that only contains themselves. In the spirit of Novelty Search (Lehman & Stanley, 2011a), this control seeks tasks that are new and different, clustering already explored tasks, and selects new tasks that do not fall into existing clusters. A similar mechanism as Algorithm 2 then partitions all tasks into interesting or boring, whereby instead of asking an autoregressive FM to predict which of the remaining tasks are boring, tasks in the same clusters are considered boring. This method is referred to as **OMNI-embed: LP + EMoI**.

We run two sets of experiments in the Crafter environment, one using repetitive tasks as boring tasks (same as the main experiments), another using repetitive and compound tasks as boring tasks (same as the experiments in Appendix Q). In light of limited compute, the experiments are run for 30 million time steps (vs. 100 million in the main experiments) and are repeated 10 times (same as the main experiments) with different random seeds. The results (presented next) show that using a pretrained sentence-embedding model as an MoI might suffice in distinguishing tasks with unique descriptions, but falls short in scenarios requiring more reasoning about the tasks, which is not captured within the pretrained embedding space.

When using repetitive tasks as boring tasks, across all metrics, the differences in performance between OMNI and OMNI-embed at 25%, 50%, 75%, and 100% of the way through training are not statistically significant (all $p > 0.05$, Mann Whitney U test). OMNI-embed's performance is comparable to that of OMNI using an autoregressive FM as MoI (Figure 29). OMNI and OMNI-embed achieve similar task success rates for each task, and induce similar patterns in task sample rates (Figure 30). This suggests that sentence-embedding models can potentially capture similar aspects of interestingness as autoregressive FMs in simple cases (but see the next paragraph where this method fails).

When using repetitive and compound tasks as boring tasks, across all metrics, the differences in performance between OMNI and OMNI-embed at 25%, 50%, 75%, and 100% of the way through training are statistically significant (all $p < 0.05$, Mann Whitney U test). OMNI using an autoregressive FM as MoI significantly outperforms OMNI-embed (Figure 31). While sentence-level embeddings may already capture a lot about task descriptions, they may not be as good as autoregressive FMs at analyzing new tasks in the context of all tasks that the agent currently performs well and poorly. For example, a sentence-level model might know that "collect wood then stone" is a different task than either "collect wood" or "collect stone", but such a model may not realize that in the context of learning, if an agent can already collect wood and stone, performing both is not a very new, interesting task with high expected learning progress vs. moving on to practicing entirely new skills. A sentence embedding model may thus overestimate the novelty of "collect wood and stone" vs. "collect coal", since the former is a compound, longer sentence, and the latter is grammatically more similar to "collect wood" (Figure 32).

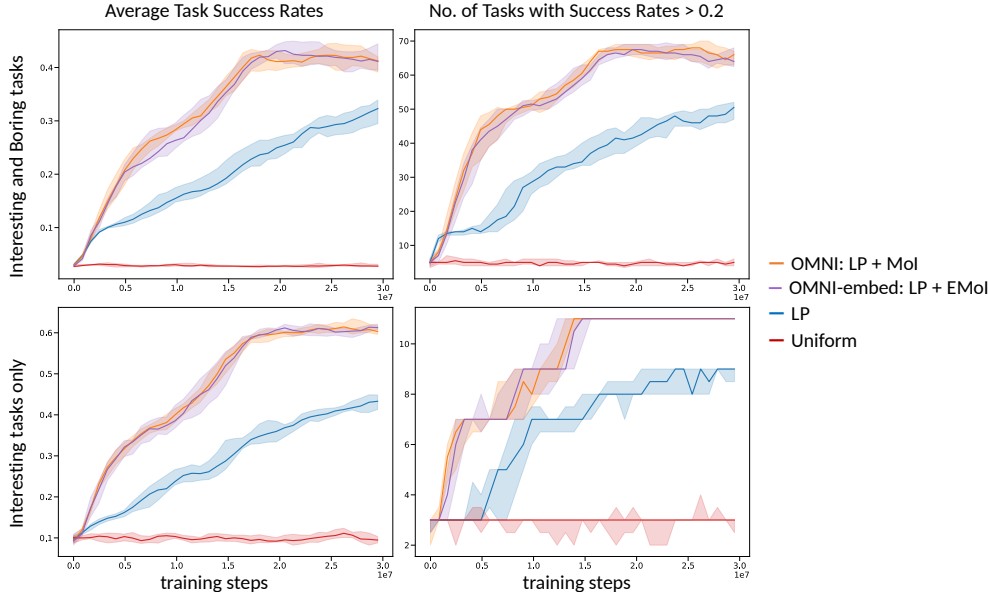

Figure 29: **Performance in Crafter on all tasks, with the only tasks considered boring being repetitive tasks.** Average task success rates and the number of tasks with success rates more than 0.2 for each method across training steps. This figure is the same as Figures 2, with an additional OMNI-embed method added for comparison. OMNI-embed achieves comparable performance to OMNI, with no statistically significant difference.

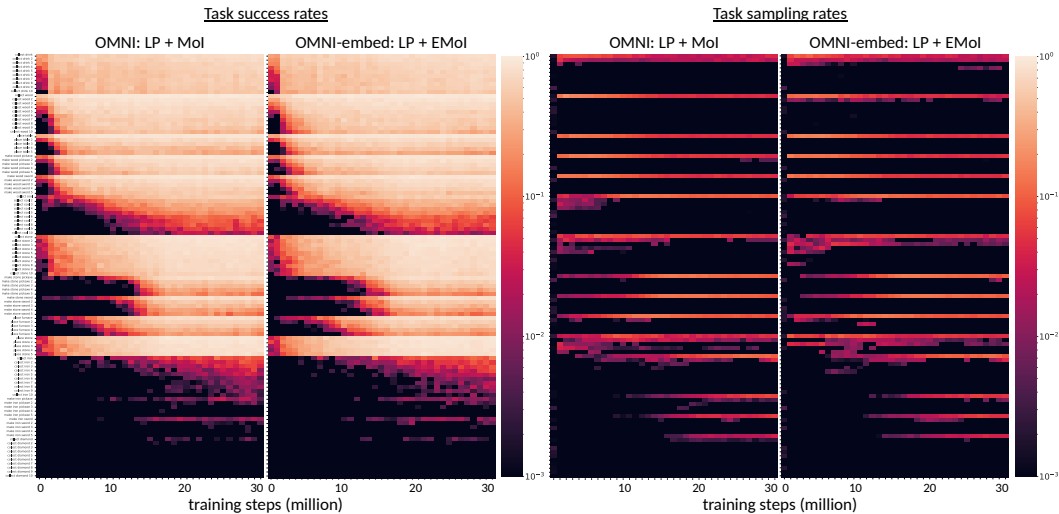

Figure 30: **(Left) Conditional success probabilities and (Right) sampling probabilities of all tasks for OMNI and OMNI-embed in Crafter.** Tasks are ordered as in Figure 2. OMNI-embed focuses on interesting tasks with high learning progress and achieves comparable task success rates to OMNI.

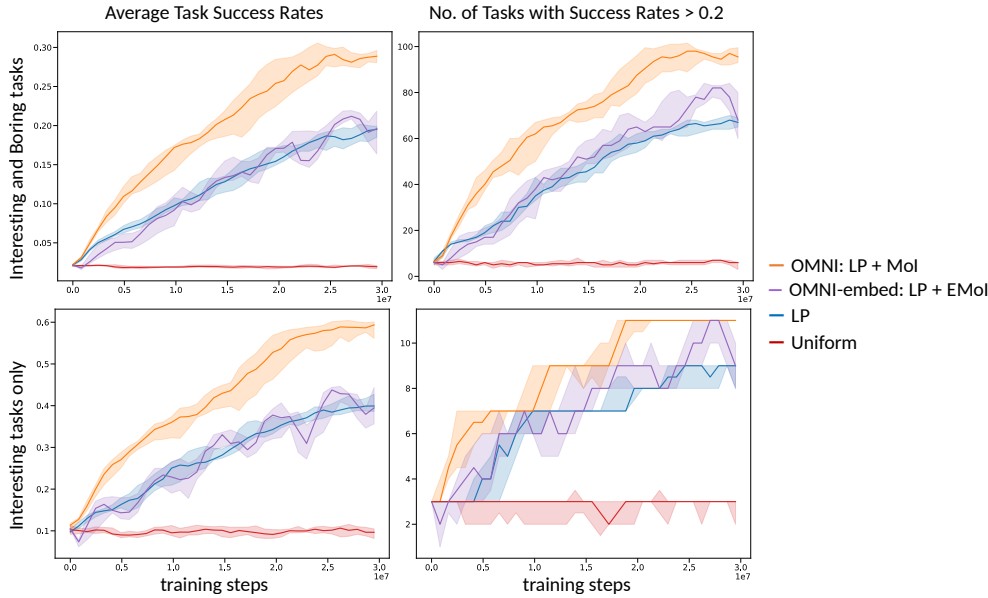

Figure 31: **Performance in Crafter on all tasks, with the tasks considered boring being repetitive and compound tasks.** Average task success rates and the number of tasks with success rates more than 0.2 for each method across training steps. Compound and repetitive tasks are those considered boring. OMNI achieves much higher average task success rates and learns more tasks than OMNI-embed.

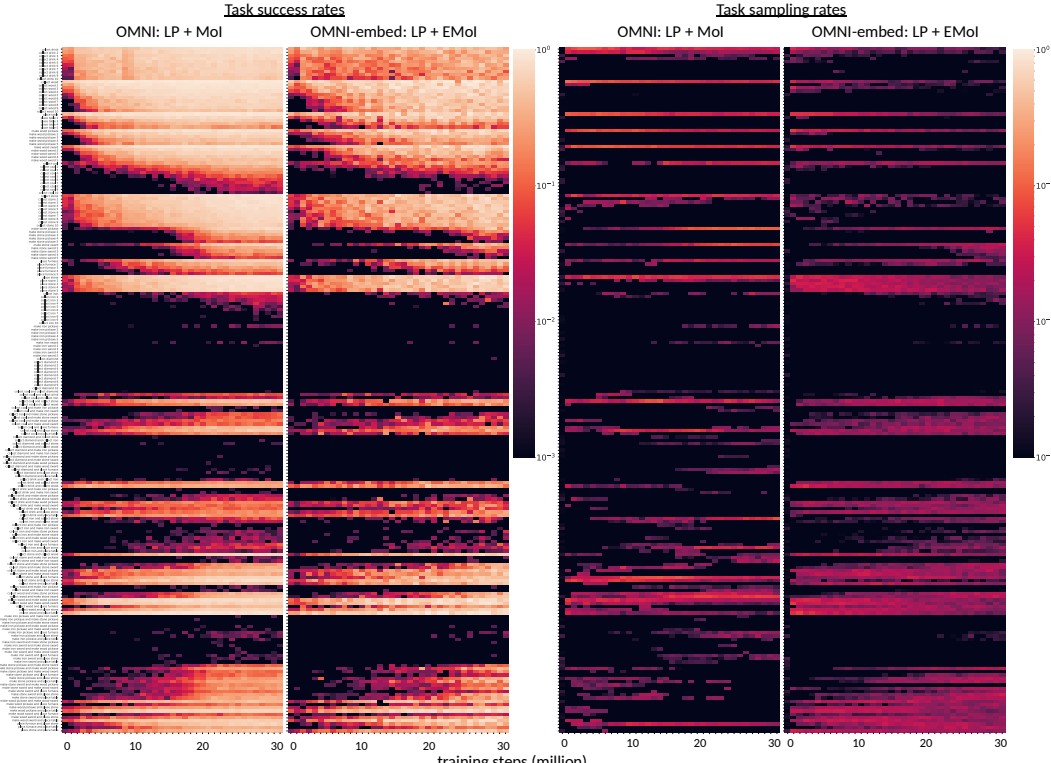

Figure 32: **(Left) Conditional success probabilities and (Right) sampling probabilities of all tasks, including compound tasks, for OMNI and OMNI-embed in Crafter.** Tasks are ordered as in Figure 20. OMNI better identifies interesting tasks and hence achieves higher task success rates than OMNI-embed.

## U  FUTURE WORK DIRECTIONS

This study explores the vision of using human notions of interestingness to accelerate open-ended learning, an approach we term *Open-endedness via Modeling human Notions of Interestingness* (OMNI). OMNI has several advantages over other methods by leveraging human concepts of interestingness to guide task selection in open-ended learning (Section 4.3). In this first version, we estimate learning progress through statistical methods and utilize a Model of Interestingness (MoI) based on human data distilled into FMs. A different version of OMNI could have the FM judge both learning progress and interestingness by giving it a history of task selection and task performance. That could allow for a more flexible notion of learning progress, as it can potentially recognize not just success or failure, but also important stepping stones and patterns leading to a solution, echoing the complexity of human learning progression. Consider an agent learning object manipulation, eventually progressing to more complex tasks like peeling an egg. Despite prolonged lack of reward, the task may not be "too difficult" as statistical learning progress might suggest. Given the agent's proven ability to handle intricate items, it may be primed for egg peeling, simply requiring more attempts. Preliminary results suggest that FMs can successfully integrate these aspects (Appendix V). Notably, as FMs continue to advance (Kaplan et al., 2020), all versions of OMNI are expected to improve correspondingly. Ultimately, OMNI offers a general recipe for accelerated learning in open-ended environments with potentially infinite tasks, and steers open-ended learning towards meaningful and interesting progress, instead of meandering aimlessly amidst endless possibilities.

Looking ahead, there are several promising avenues for future work. One possibility is to incorporate multi-modal models, such as vision-language models and other modalities into the *MoI*. This could give the MoI richer representations and a better comprehension of the agent's capabilities, facilitating a more accurate assessment of the agent's learning progress and task diversity. For example, a vision-language MoI might see that the agent is making progress on or very close to solving a task for which it is getting no reward, such as peeling an egg. Another idea is to allow the MoI to autonomously analyze quantitative performance measures, make its own assessment of learning progress, and incorporate that into its notion of interestingness. Our preliminary investigations indicate that FMs, such as GPT-3 and GPT-4, can automatically analyze numerical results and adjust their understanding of interestingness (Figures 24 and 27).

Critically, by Goodhart's law, we expect that any model will have pathologies uncovered once it is a target metric being optimized against. For instance, fine-tuning a task generator to produce tasks that the MoI finds interesting might eventually result in the MoI not being a good indicator of what is interesting. Hence, refining and updating the MoI with additional human feedback could lead to more effective learning systems, an algorithm within the OMNI paradigm we call *Open-Endedness with Human Feedback* (OEHF). Similar to Reinforcement Learning with Human Feedback (RLHF) (Christiano et al., 2017), the objective is to train a model that can effectively capture an ineffable property that, although challenging to quantitatively measure, is readily identifiable upon observation (e.g., a backflip for RLHF, or whether a task is interesting for OEHF). One version would be to build upon the insights of this paper and start with a model *already* well-versed with human concepts of interestingness (via unsupervised pre-training on internet-scale human data), and further fine-tune that MoI with additional human evaluation of its output (as often as necessary). Such fine-tuning can help minimize suboptimal interestingness judgements, enhance the MoI's understanding of skills not initially present in its zero-shot repertoire, or tailor the MoI to a specific domain.

## V  INTEGRATING MORE ASPECTS OF INTERESTINGNESS INTO FMS

For future research, we plan to explore the potential of leveraging FMs to address multiple aspects of interestingness, including learning progress. By examining these aspects individually or in a combined manner, we aim to develop more robust and adaptive models that can efficiently cater to a wide range of tasks and applications. Preliminary evidence suggests that FMs, such as GPT-4, possess the capacity to integrate multiple aspects of interestingness (Figure 33). Further investigation and experimentation in this direction could lead to the development of AI models that exhibit a more sophisticated approach to learning by effectively capturing and integrating various aspects of interestingness.

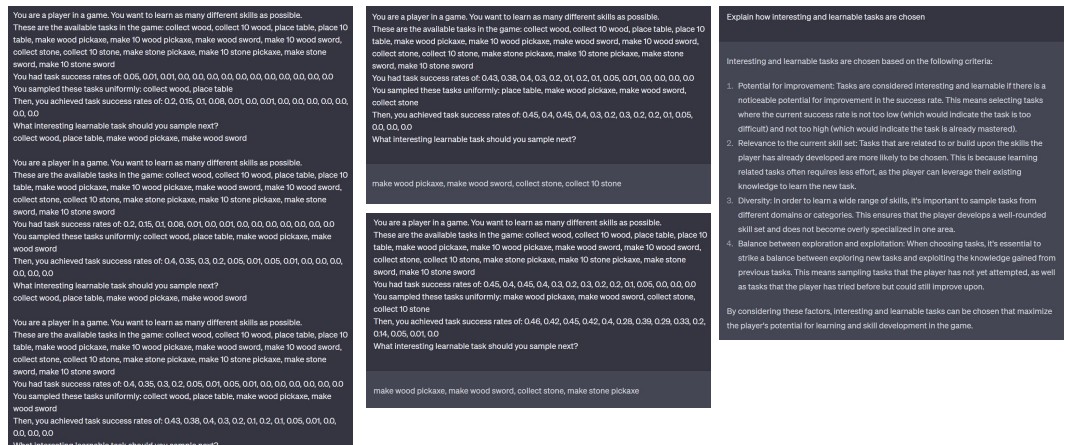

Figure 33: (**Left**) Few-shot examples part of first prompt input to GPT-4. (**Middle**) GPT-4's prediction of learnable and interesting tasks. (**Right**) GPT-4 explanation of how interesting and learnable tasks are chosen. The darker shaded area is the input prompt, the lighter shaded area is the generated output. By providing GPT-4 with a few examples of what learnable and interesting tasks are, the model can synthesize these concepts and consistently generate a set of tasks that meet the desired criteria.

# W    TENSIONS BETWEEN OPEN-ENDEDNESS AND SAFETY

Open-ended algorithms have tremendous potential to unlock unbounded creativity and catalyze scientific discoveries, including advancing AI research (Nguyen et al., 2015; Clune, 2019; Stanley et al., 2023). However, this uncharted expanse also harbors unique safety challenges, as the inherently unpredictable nature of such algorithms can lead to outcomes misaligned with human values and expectations (Ecoffet et al., 2020; Clune, 2019). Recognizing this, it remains an open research question how to explore and take advantage of such algorithms in a safe, value-aligned way. One method could be to use human feedback to update the model of interestingness, similar to methods used in RLHF (Christiano et al., 2017; Ding et al., 2023). Another method could be to use AI feedback to update the model of interestingness (Bai et al., 2022; Bradley et al., 2023), minimizing the chance of selecting potentially dangerous tasks. Further research is required to understand the degree to which such methods will work, and to invent new, better methods, so that we can shepherd open-ended algorithms towards beneficial and secure applications, safeguarding against unintended consequences while embracing their transformative potential.

