# OpenReview forum: "OMNI: Open-endedness via Models of human Notions of Interestingness"
_ICLR.cc/2024/Conference — ICLR 2024 poster_

### Official Review · Reviewer_CZLW · 2023-10-30

**Soundness:** 3 good
**Presentation:** 3 good
**Contribution:** 2 fair
**Rating:** 8
**Confidence:** 3

**Summary:**

Paper introduces a new curriculum generation approach for reinforcement learning agents. From the high level, it includes 1) an LP model, which ranks the tasks based off the difficulty of the current learning progress of the agents; 3) an interestingness model (Mol), which ranks the tasks out of the "interestingness", a subjective view by human infused to the LLM. The proposed method, OMNI, is evaluated on life-long RL in two domains: crafter and babyAI. Basically, the agent will follow the task proposed by OMNI, an ablative version of OMNI, or a random task selector. The results show that agents learned with the curriculum produced by OMNI have clear advantages.

**Strengths:**

+The research topic presented here is relevant and important. There have been active explorations of how to turn the powerful LLM into general agents. Building a curricula generator is definitely very promising and should be of interest to a range of audiences from general agents to LLMs.

+The paper is clear and well-written. The proposed method is technically sound -- LLM indeed has some infused human prior that could help determine the "interestingness" of the tasks and help build more reasonable curricula for RL.

+The results on crafter and babyAI look quite promising. The advantages over random curriculum generators and ablative versions of OMNI are clear. The additional experiments in the appendix are very thorough.

**Weaknesses:**

At this point, I do not have major concerns. But I do hope the authors can help address the following questions:

-What if the LLM does not have the human prior or reasoning capacity about the task "interestingness" on the given domain?

-Based on my personal experiences of training agents in Crafter, an agent with a random task generator (Uniform) should not be that bad as it should be able to master not just a limited set of tasks as the curve in Figure 3 suggested. Can the authors clarify this?

-If my understanding is correct, LP seems to be more crucial to the success of mastering more challenging tasks as it takes the learning process into consideration, while "interestingness" is about not wasting time on tasks that could not lead to meaningful learning. Therefore, if an unlimited round of learning is allowed, an agent with LP only should ultimately match the results of full OMNI on the average success rate. However, this is not the case in all the experiments. Maybe the LP baseline is not run for enough rounds? I would like to learn more about your thoughts on this.

Minor: some references on building open-ended agents in open-world environments should be cited [1-5]

[1] open-world control: https://arxiv.org/abs/2301.10034

[2] DEPS: https://arxiv.org/abs/2302.01560

[3] STG transformer: https://arxiv.org/abs/2306.12860

[4] Plan4MC: https://arxiv.org/abs/2303.16563

[5] GITM: https://arxiv.org/abs/2305.17144

**Questions:**

See "weaknesses"

---

> ### Author Response · Authors · 2023-11-17
>
> Thank you for your very insightful review, and kind words about the paper's strengths. We now address each of your concerns and questions.
>
> > What if the LLM does not have the human prior or reasoning capacity about the task "interestingness" on the given domain?
>
> Given the extensive dataset that LLMs have been trained on, it is unlikely that LLM would not already have some prior on any task that is human-generated. Regardless, we believe OMNI being helpful on many/ most tasks is still valuable, even if it is not helpful on all.
>
> In the case where the LLM does not have a good enough notion of interestingness, we propose to use human feedback to train (or fine-tune) a model of interestingness. This is termed Open-Endedness with Human Feedback (OEHF) in Section 6 (Section 5 of the original manuscript), as part of future work. Similar to Reinforcement Learning with Human Feedback (RLHF), one could potentially train a model that can effectively capture an ineffable property that, although challenging to quantitatively measure, is readily identifiable upon observation (e.g., a backflip for RLHF, or whether a task is interesting for OEHF).
>
> > Based on my personal experiences of training agents in Crafter, an agent with a random task generator (Uniform) should not be that bad as it should be able to master not just a limited set of tasks as the curve in Figure 3 suggested. Can the authors clarify this?
>
> If all of the tasks are interesting (i.e., only the 15 tasks designed in the original Crafter environment), using uniform sampling would be sufficient. This effectiveness can be attributed to the fact that these environments are often already designed with some human intuition for interestingness, only including tasks or features that we instinctively find interesting.
>
> However, in the broader context of open-ended learning where we want environments with infinite possible tasks, it is extremely difficult to design an environment or task space that only contains interesting features. Take a kitchen environment for example, there can be endless variations of possible tasks (e.g., putting a cup in slightly different positions), and it is almost impossible to curate a set of tasks that are all interesting. We modified Crafter and BabyAI environments by expanding the task space beyond the carefully selected few in order to emulate this phenomenon from vast, open-ended environments. It is because there are many boring or impossible tasks in the modified versions of these domains that uniform sampling fails.
>
> In new experiments, we demonstrate OMNI’s ability to handle just such an infinite task space (the set of all tasks specifiable in natural language). Please see our general reviewer response for a summary.
>
> > If my understanding is correct, LP seems to be more crucial to the success of mastering more challenging tasks as it takes the learning process into consideration, while "interestingness" is about not wasting time on tasks that could not lead to meaningful learning. Therefore, if an unlimited round of learning is allowed, an agent with LP only should ultimately match the results of full OMNI on the average success rate. However, this is not the case in all the experiments. Maybe the LP baseline is not run for enough rounds? I would like to learn more about your thoughts on this.
>
> In an ideal scenario with infinite compute, an agent guided solely by LP could converge to performance levels of an agent utilizing the full OMNI approach. However, in reality, having infinite compute is impossible. In a truly open-ended setting, there are an infinite number of uninteresting tasks that LP will inevitably get distracted by. For example in the real world, there can be endless variations of mathematical, or even addition, questions (e.g. just by changing constants) that the agent could attempt. In the Crafter environment, LP is distracted by boring repetitive tasks, which require substantial compute and training steps, whereas OMNI is able to filter out the boring tasks and only focus on the interesting tasks to gain more meaningful learning.
>
> > Minor: some references on building open-ended agents in open-world environments should be cited [1-5]
>
> Thank you for sharing these works, we have included them in the Related Works section of the revised manuscript.
>
> We hope we have answered all your questions and concerns. We fear the current scores will preclude our paper from being published and shared with the ICLR community, despite the many positive reviewer comments on its originality and likely long term impact. If you are open to considering a higher score, that would improve the chances of the paper being accepted and we would deeply appreciate it.

---

> > ### Comment · Reviewer_CZLW · 2023-11-21
> >
> > Thank you. My concerns have been addressed. I've raised my score accordingly.

---

### Official Review · Reviewer_LSLH · 2023-11-01

**Soundness:** 2 fair
**Presentation:** 4 excellent
**Contribution:** 3 good
**Rating:** 6
**Confidence:** 5

**Summary:**

This paper focuses on an interesting and practical scenario that the agent is exploring in an open-ended environment and learning new behaviors forever. Massive unlearned tasks exists in such environment, and how to choose the task to learn becomes a challenge. Beyond conventional choosing criteria, interestingness is an important aspect which will be largely considered by humans when exploring the open-ended environment, but is hard to be measured. This paper proposes to use large models, which already have encoded many human knowledge, as a model of interestingness, providing guidance for the agent to choose the task to learn. The authors further provide a simple implementation for such the above idea. Two environments (Crafter and BabyAI) are tested in experiments, and good results are obtained for both environments.

**Strengths:**

This paper targets an important problem. Open-endedness needs to be considered in many practical scenarios, where agents need to learn and improve themselves in interactive environments.

The intuition and idea behind this paper is very reasonable and promising. Large models offer us many potentials, making some previously impossible tasks feasible now, and this paper successfully finds such a task. Interesting tasks are hard to define, and now with the help of large models, one can ask what tasks are interesting and explore environments like a human. Also, finding interesting tasks is very important in open-endedness research.

Overall, I like the idea of this paper, and believe that this principle will have impact on open-endedness research.

**Weaknesses:**

The baselines compared in this paper are not strong enough. All baselines are intuitively set, and some previously proposed methods (some are discussed in related work, like curriculum learning for RL) need to be included.

Although the authors claim that OMNI is a principle and this paper only presents one instance, practical implementation is essential to verify the proposed principle, and some designs in the current algorithm may need to be improved.
- It might not be practical to know all candidate tasks in advance, and just let the large model choose one. In the RL setup, the agent needs to explore the environment and finds out all candidate tasks.
- The downsampling method for boring tasks is quite simple now (multiplying by 0.001). The applicability of this method needs to be discussed and analyzed.

Human alignment also needs to be discussed. This paper just uses the interestingness of large models as that of human to do exploration. On the other hand, alignment is an unsolved question that targets aligning human and large models and avoiding dangerous actions. Methods like RLHF are proposed here. Considering that human alignment is not perfect now, how the interestingness of large models be used for choosing tasks and some important aspects, like safety, are also ensured at the same time?

The missing related work (https://arxiv.org/pdf/2302.06692.pdf) uses language model to suggest plausibly useful goals, which shares similar idea with this paper.

**Questions:**

Why is the survival component removed in the Crafter environment? This is quite important in a practical open-ended environment. Without considering the survival component, the aspects that need to be considered during choosing the task become less adequate and complex, which goes against the motivation of using LLMs here.

In experiments, how does the large model know what tasks are interesting tasks, given that interesting tasks are pre-defined by the authors?

It’s better to show instructions used in determining interesting tasks. How to query the large model and what kind of results are obtained from it? An example here might be helpful.

---

> ### Author Response · Authors · 2023-11-17
>
> Thank you for your feedback. We appreciate your recognition of the strengths in our approach and presentation. We now address each of your concerns and questions.
>
> > The baselines compared in this paper are not strong enough. All baselines are intuitively set, and some previously proposed methods (some are discussed in related work, like curriculum learning for RL) need to be included.
>
> Learning progress (LP) is an RL curriculum learning method, which has been demonstrated by previous works as a strong baseline (Kanitscheider et al., 2021). Using LP as an RL curriculum, Kanitscheider et al. (2021) could train Minecraft agents to collect diamonds from scratch, which has been recognized as an extremely challenging RL task. We will make this clearer in the manuscript.
>
> We have added more experiments to our study. These new experiments, like related works (e.g., intrinsic motivation, novelty search, diversity), rely on pre-specified definitions of what counts as interestingly different. We compare OMNI with an alternative MoI which relies on predefined heuristics based on sentence embeddings. In this alternative, embedding-based MoI (eMoI), interestingness is determined by the distances between sentence embeddings of task descriptions. We show where such predefined metrics fail, and that OMNI outperforms the controls significantly. Please see Appendix P in the revised manuscript for the full details and results.
>
> > It might not be practical to know all candidate tasks in advance, and just let the large model choose one. In the RL setup, the agent needs to explore the environment and finds out all candidate tasks.
>
> Thank you for pointing this out. We also knew that such a domain was the next frontier for OMNI (we planned on it being a follow-up paper, but are happy to add it to this paper instead). So immediately after submitting the original version, we began testing OMNI in an infinite, complex, robotics domain, and are pleased to report that OMNI far outperforms controls (the set of all tasks expressible in natural language). The revised manuscript includes these results, please see our general reviewer response for a summary.
>
> > Human alignment also needs to be discussed. [...] Considering that human alignment is not perfect now, how the interestingness of large models be used for choosing tasks and some important aspects, like safety, are also ensured at the same time?
>
> We agree that human alignment and safety are important and interesting research challenges for open-endedness. We have added a discussion on this topic in Appendix R of the revised manuscript.
>
> > The missing related work (https://arxiv.org/pdf/2302.06692.pdf) uses language model to suggest plausibly useful goals, which shares similar idea with this paper.
>
> Du et al. (2023) was discussed in Section 2.3 of Related Works in the original manuscript. Please let us know if there are more comments regarding it that you feel we should add.
>
> > Why is the survival component removed in the Crafter environment? This is quite important in a practical open-ended environment. Without considering the survival component, the aspects that need to be considered during choosing the task become less adequate and complex, which goes against the motivation of using LLMs here.
>
> By removing the survival component, task completion is less susceptible to external stochastic events like whether a skeleton has killed the player. It removes the need for the agent to learn and continually apply survival tactics against enemies or for food gathering. In other words, it allows us to focus on evaluating the tasks specified with less evaluation noise.
>
> For completeness, based on your request, we conducted new experiments with the original Crafter environment, where survival components are retained. The findings from these experiment setup are consistent with our initial results, where OMNI outperforms Learning Progress and Uniform sampling significantly. For more details and results, please see Appendix O of the revised manuscript.
>
> > In experiments, how does the large model know what tasks are interesting tasks, given that interesting tasks are pre-defined by the authors?
>
> There is a set of “interesting tasks” that we manually chose, but that set of tasks is only for purposes of evaluating the trained model. The OMNI algorithm is not made aware of these choices. In OMNI, we tell the LLM which tasks are considered done well by the agent, and prompt the LLM to predict whether the remaining tasks are interesting or not.
>
> > It’s better to show instructions used in determining interesting tasks. How to query the large model and what kind of results are obtained from it? An example here might be helpful.
>
> Full LLM prompts are provided in Appendices B and C. Appendix F (Appendix D in original manuscript) showcases a step-by-step example of how the model declares tasks as interesting or not. We have also added pseudocode in Section 3 of the revised manuscript for additional clarity.

---

> > ### Author Response · Authors · 2023-11-22
> >
> > Dear reviewer,
> >
> > Two reviewers have substantially increased their scores upon seeing the rebuttal responses and additional experiments, including one experiment on an entirely new, complex, robotics domain that also (critically) has an infinite task space. We believe we have addressed all of your concerns in our rebuttal above. We would deeply appreciate it if you could take a look and see if you might consider increasing your score?
> >
> > Thank you.

---

### Official Review · Reviewer_qXzy · 2023-11-03

**Soundness:** 3 good
**Presentation:** 2 fair
**Contribution:** 3 good
**Rating:** 8
**Confidence:** 4

**Summary:**

The paper introduces a novel approach to enhance open-ended learning algorithms, which aim to continually learn new and interesting behaviors in vast environments with innumerable tasks. The core challenge addressed is the existing inability to quantify and prioritize tasks that are not only learnable but also inherently interesting—deemed to be significant due to their worth and novelty. The authors propose the concept of Open-endedness via Models of human Notions of Interestingness (OMNI), leveraging large language models (LLMs) to serve as models of interestingness (MoIs). These LLMs are presumed to embody human perceptions of interestingness, having been trained on extensive human-generated data. The paper demonstrates through experiments that OMNI can direct open-ended learning towards tasks that are both learnable and interesting, showing superiority over baselines that rely solely on uniform task sampling or learning progress.

**Strengths:**

- The idea of leveraging large language models to approximate human notions of interestingness in open-ended learning environments is novel and holds substantial promise.
-  The paper includes detailed experiments across two domains that showcase the effectiveness of the OMNI framework in focusing learning on tasks that are both interesting and learnable.

**Weaknesses:**

## Main concerns
- The major weakness of this work is the lack of comprehensive comparitive analysis. The authors mention and properly cite a number of curriculum learning papers, and also works attempting to quantify interestingness. Why isn’t any of these methods used for comparison? It is very obvious that OMNI would outperform uniform sampling and LP. The authors should try out LP + other metrics of interestingness, as described in Section 2.2.

- The performance of OMNI and other baselines on BabyAI is extremely bad (success rate < 0.008). Have the authors taken a look at the policies and understood what is happening? I understand this includes hard tasks, but have the authors considered plotting the results for only interesting tasks, similar to Crafter?

- In the experiments, the authors use a pre-defined set of interesting, boring and challenging tasks. However, it would also be interesting to try the methods in a more open-ended setting whereby they use a random task generator that procedurally generates virtually unlimited number of tasks.

- The evaluation of the trained models is performed only on the previously seen tasks. I was wondering if the authors considered evaluating on held-out, and particularly out-of-distribution (OOD) tasks to assess the robustness of the trained methods.

## Content and Writing Style Issues
- A disproportionate emphasis on potential implications and future work in the paper might obscure the immediate contributions. For example, the Conclusion is > 1.25 page, mostly discussing future work. While this is very interesting, it feels to me that it takes the space from what actually was important in this paper and was moved to the Appendix. In the end, the reader has to go back and forth from main part of the paper to appendix to follow the ideas described in these different sections. It is my recommendation that the authors carefully consider what they want to include in the main part of the paper vs the appendix.

> "Perhaps, someday, unlimited energy and a cure for cancer can be added to the list", "ephemeral fuel ...", "eons of human experience", "shadows of what human sense could do", "intriguing prospect has arisen", "vast troves...", "to borrow from Newton",...
- Additionally, the stylistic choice of using lyrical language is atypical for scientific papers and could compromise the accessibility and clarity for a wider audience.

> "OMNI has the potential to significantly enhance the ability of AI to intelligently select which tasks to concentrate on next for endless learning and could pave the way for self-improving AI and AI-Generating Algorithms"
-  I think there is a giant gap between the what the authors presented in this work, compared to what they claim it has the potential of doing. I would recommend removing/rewarding such statements.

**Questions:**

See above.

UPDATE: I have updated the score following the rebuttal.

---

> ### Author Response · Authors · 2023-11-17
>
> Thank you for your thoughtful and constructive feedback. We appreciate your positive remarks and critical analysis of our work. We now address each of your concerns and questions.
>
> > The major weakness of this work is the lack of comprehensive comparitive analysis. [...] The authors should try out LP + other metrics of interestingness, as described in Section 2.2.
>
> As per your suggestion, we have added more experiments to our study. The new experiments, like the methods in section 2.2 (e.g., intrinsic motivation, novelty search, diversity), rely on pre-specified definitions of what counts as interestingly different. We compare OMNI with an alternative MoI which relies on predefined heuristics based on sentence embeddings. In this alternative, embedding-based MoI (eMoI), interestingness is determined by the distances between sentence embeddings of task descriptions. Please see Appendix P in the revised manuscript for the full details and results.
>
> We show that the embedding-based MoI (eMoI) might suffice in distinguishing tasks with unique descriptions, but falls short in scenarios requiring more reasoning about the tasks. This exemplifies where pathologies might happen when we attempt to quantify the ineffable notion of interestingness. The additional results further reinforce the advantages of OMNI in focusing on learnable and interesting tasks, and highlights where autoregressive LMs might have an edge over predefined metrics of interestingness.
>
> >The performance of OMNI and other baselines on BabyAI is extremely bad (success rate < 0.008). Have the authors taken a look at the policies and understood what is happening? I understand this includes hard tasks, but have the authors considered plotting the results for only interesting tasks, similar to Crafter?
>
> As you guessed, the average success rate is low because it is an average across the entire taskset of 1364 tasks. Most tasks are very difficult, which are not even completed at a success rate of 0.05 at any time step by any method. The reason why the tasks are so difficult in BabyAI is that each task does not depend on the target object(s) or environment configuration in that episode. For example, the target object distribution for a GoTo task can be any object in the environment distribution. If the object is in another locked room, the GoTo task would implicitly require the agent to find a key, unlock and open the door, before navigating to the target object.
>
> We like your suggestion of plotting on interesting tasks only. We did a version of that by averaging success rates over tasks with instruction lengths of one or two only (see additional Figure 15 in Appendix K). OMNI significantly outperforms controls, and shows a more impressive completion rate of 16%.
>
> > In the experiments, the authors use a pre-defined set of interesting, boring and challenging tasks. However, it would also be interesting to try the methods in a more open-ended setting whereby they use a random task generator that procedurally generates virtually unlimited number of tasks.
>
> Thank you for the suggestion. We are equally as excited to share the results of OMNI in an infinite, complex, robotics domain, and are pleased to report that OMNI far outperforms controls (the set of all tasks expressible in natural language). The revised manuscript includes these results, please see our general reviewer response for a summary.
>
> > The evaluation of the trained models is performed only on the previously seen tasks. I was wondering if the authors considered evaluating on held-out, and particularly out-of-distribution (OOD) tasks to assess the robustness of the trained methods.
>
> In the current version, we are not claiming that OMNI can generalize to OOD tasks by training on interesting and learnable tasks. We agree that testing this would be an interesting future research direction, especially once OMNI is trained at scale.
>
> > A disproportionate emphasis on potential implications and future work in the paper might obscure the immediate contributions. [...] While this is very interesting, it feels to me that it takes the space from what actually was important in this paper and was moved to the Appendix. In the end, the reader has to go back and forth from main part of the paper to appendix to follow the ideas described in these different sections.
>
> We apologize for the need to switch between the main text and the appendix. If accepted to ICLR, we will commit to moving some of the experimental details back to the main text, and some future work to the appendix.
>
> > the stylistic choice of using lyrical language is atypical for scientific papers and could compromise the accessibility and clarity for a wider audience. I think there is a giant gap between the what the authors presented in this work, compared to what they claim it has the potential of doing. I would recommend removing/rewarding such statements.
>
> We have toned down some of the writing style and claims in the revised manuscript.

---

> ### Comment · Reviewer_qXzy · 2023-11-21
> **Response to rebuttal**
>
> I thank the authors for a detailed response to my concerns and for adding new additional experiments that certainly improve the quality of the manuscript. I have updated my score accordingly.

---

### Official Review · Reviewer_MxZx · 2023-11-04

**Soundness:** 2 fair
**Presentation:** 1 poor
**Contribution:** 2 fair
**Rating:** 3
**Confidence:** 4

**Summary:**

This paper proposes to use LLMs to measure how interesting tasks are for open-ended algorithms to learn. The problem which they tackle is that for open-ended algorithms, there are an infinite number of tasks which that agent could try to learn. This can be problematic for two reasons: the tasks may be unlearnable, or they may be uninteresting. The paper notes that the notion of "interestingness" is hard to measure exactly, but that humans know it when they see it. The paper proposes to combine measures two measures: learning progress (to ignore tasks which are unlearnable, and which has been explored in previous work), and interestingness (which is here measured using an LLM). They construct variants of the Crafter and BabyAI environments to test this approach, and find that it outperforms uniform sampling of tasks and prioritizing tasks by learning progress alone.

Overall, this paper addresses an interesting problem and I think the approach has potential, but I do not think it's ready for publication due to the experiments which are not convincing enough and the presentation which needs work. I think this paper would benefit from another revision cycle giving time to add experiments on more convincing environments and improve the writing, after which it would make a strong submission.

**Strengths:**

- The paper is well motivated and the problem is interesting. They correctly identify one of the fundamental issues with open-ended learning algorithms, i.e. that if the agents choose tasks based on novelty, feasibility or information gain alone, they may end up choosing tasks which are completely uninteresting to humans, but which a  _tabular rasa_ agent has no reason to deprioritize.

**Weaknesses:**

- Unfortunately, the experimental section of the paper does not live up to the ambitious ideas described in the intro. Only two (fairly simple) environments are considered (Crafter and one MiniGrid task), and the experiments feel contrived. For Crafter, "Impossible" tasks are generated by simply assigning the agent 0 reward. "Boring" tasks are generated by asking the agent to repeat the same task several times. Because of this, it is possible to solve the Crafter setup with very simple heuristics, and an LLM is not really needed. Granted, LLMs are more general, but it would be a lot more compelling if this method were evaluated on tasks which are not solvable with simple heuristics.
- The paper's presentation needs improving. The paper is missing a lot of details about the algorithm. While Figure 1 gives a high-level overview of the system, it is still fairly vague and the rest of the description is all in the text and not very precise. It would be helpful to have the entire algorithm spelled out in pseudocode (in the main text), the exact process by which tasks are chosen described in equations or pseudocode, etc.

**Questions:**

Here are my questions and suggestions:

- My main suggestion for improving this paper is to test on one or more challenging environments. While artificial settings like the ones used in this paper are nice for proof of concepts or ease of interpretation, they need to be complemented by experiments on a more challenging domain where simple heuristics are insufficient. I would suggest trying this either on Minecraft, which has been used for open-ended research [1], or NetHack, which is another very complex open-ended game where the extrinsic reward is insufficient (it is also fast to run) [2].
- My second suggestion is improving the presentation (see my comments above). It's also not clear what criteria the agent uses to decide it can do a task "well" - this this based on success rate or reward? Please clarify. In addition to this, I think the Conclusion/Discussion/Future Work section is too long at over 1 page. This saved space can be used for extra experiments in this paper.



[1] Voyager: An Open-Ended Embodied Agent with Large Language Models (Wang et al)

[2] The NetHack Learning Environment (Kuttler et al, NeurIPS 2020)

---

> ### Author Response · Authors · 2023-11-17
>
> Thank you for your thoughtful review. We greatly appreciate your acknowledgement of the novelty and potential of OMNI. We now address each of your concerns and questions.
>
> > Unfortunately, the experimental section of the paper does not live up to the ambitious ideas described in the intro. Only two (fairly simple) environments are considered (Crafter and one MiniGrid task), and the experiments feel contrived. For Crafter, "Impossible" tasks are generated by simply assigning the agent 0 reward. "Boring" tasks are generated by asking the agent to repeat the same task several times. Because of this, it is possible to solve the Crafter setup with very simple heuristics, and an LLM is not really needed. Granted, LLMs are more general, but it would be a lot more compelling if this method were evaluated on tasks which are not solvable with simple heuristics.
> > My main suggestion for improving this paper is to test on one or more challenging environments. While artificial settings like the ones used in this paper are nice for proof of concepts or ease of interpretation, they need to be complemented by experiments on a more challenging domain where simple heuristics are insufficient. I would suggest trying this either on Minecraft, which has been used for open-ended research [1], or NetHack, which is another very complex open-ended game where the extrinsic reward is insufficient (it is also fast to run) [2].
>
> It is imperative to note that transitioning to such environments entails significant challenges. If we insist that the first paper of every new idea be showcased on a very complex domain, (1) we would really disadvantage academic labs, who do not have the compute resources often required for such complex domains, and (2) require many years of work before a student can publish their first paper.
>
> Voyager (Wang et al., 2023) was done by a team of full-time scientists from NVIDIA, and probably has orders of magnitudes more compute than any academic lab. We purposefully chose Crafter, the academic equivalent of Minecraft, because it has similar dynamics but is within budget for an academic lab. Furthermore, both Crafter and MiniGrid have been shown to be challenging RL domains (Hafner, 2021; Du et al., 2023; Chevalier-Boisvert et al., 2018; Mu et al, 2022).
>
> Nonetheless, there is value in extending OMNI to a more complex, open-ended domain. Hence, our revision includes additional results on a very complex, open-ended, realistic embodied kitchen robotics domain, AI2-Thor (Kolve et al., 2017). Please see our general reviewer response for a summary.
>
> > The paper's presentation needs improving. The paper is missing a lot of details about the algorithm. While Figure 1 gives a high-level overview of the system, it is still fairly vague and the rest of the description is all in the text and not very precise. It would be helpful to have the entire algorithm spelled out in pseudocode (in the main text), the exact process by which tasks are chosen described in equations or pseudocode, etc.
>
> We like your idea of including pseudocode for the entire algorithm, and have included it in Section 3 of the revised manuscript. Furthermore, as promised at the start of the Appendix, we will open-source the code to provide full transparency.
>
> > My second suggestion is improving the presentation (see my comments above). It's also not clear what criteria the agent uses to decide it can do a task "well" - this this based on success rate or reward? Please clarify. In addition to this, I think the Conclusion/Discussion/Future Work section is too long at over 1 page. This saved space can be used for extra experiments in this paper.
>
> Appendix F (Appendix D in original manuscript) explains how the agent decides which tasks are done well. On top of the details in the original manuscript, we hope that the additional pseudocode clarifies this. Reviewer LSLH indicated that the presentation was excellent and Reviewer CZLW commented that “the paper is clear and well-written”. Given the additional updates, we hope you find the presentation better. If accepted to ICLR’24, we will commit to moving some of the experimental details back to the main text, and move some future work to the appendix. If there are any other missing details, please let us know and we will add them to the manuscript.
>
> Your score is substantially lower than the other reviews, and if not changed likely will lead to the paper not being published. Are our improvements enough for you to consider increasing your score? We deeply appreciate your consideration.

---

> > ### Author Response · Authors · 2023-11-22
> >
> > Dear reviewer,
> >
> > Two reviewers have substantially increased their scores upon seeing the rebuttal responses and additional experiments, including one experiment on an entirely new, complex, robotics domain that also (critically) has an infinite task space. We believe we have addressed all of your concerns in our rebuttal above. We would deeply appreciate it if you could take a look and see if you might consider increasing your score?
> >
> > Thank you.

---

### Author Response · Authors · 2023-11-17

Thank you for reviewing our work. We are delighted that “the paper is well motivated and the problem is interesting” (MxZx), “relevant” (CALW), and “targets an important problem” (LSLH); that “the paper is clear and well-written” (CZLW); that “the proposed method is technically sound” (CZLW), “novel and holds substantial promise” (qXzy); that the experiments “showcase the effectiveness of the OMNI framework” (qXzy), are “detailed” (qXzy) and “very thorough” (CZLW). Finally, we are happy to see that this work “should be of interest to a range of audiences from general agents to LLMs” (CZLW), and “that this principle will have impact on open-endedness research” (LSLH).

We have attempted to address all concerns, which has significantly improved the manuscript. **Most importantly, we have added a new results section in an entirely new, very complex domain that has an infinite task space (the set of all tasks describable in natural language), which addresses a main concern of reviewers. The results in this much more challenging, realistic domain confirm all of the previous results and claims.**

In the revised manuscript, we have highlighted the changes and specified the reviewer to whom each change is addressed. We feel this paper makes an important, helpful contribution (i.e., identifying a core problem in open-ended learning and proposing a scalable solution) that the ML community will benefit from if published. The current scores are low and will likely prevent publication. We hope you will keep an open mind to changing your scores in light of our improvements. We humbly ask if you are willing to increase your score.

# The prominent changes are as follows:
1. Results of OMNI in an embodied kitchen domain (AI2-Thor), which is more complex, open-ended, and has infinite task space. All of which confirm the principles of OMNI in an entirely new, and significantly more challenging domain.
2. Several additional experiments, more exhaustively analyzing the performance of OMNI in different experiment setups and using alternative metrics of interestingness. All additional experiments show results that validate the design choices and principles behind OMNI.
3. Additional pseudocode for the entire OMNI algorithm.
4. More discussion on potential human alignment issues.
5. We have additionally responded to each reviewer comment individually (see reviewer-specific replies) and based on your feedback improved the writing throughout.

Your feedback has enabled us to substantially improve the paper. With these changes, we feel the work is suitable to be accepted to ICLR 2024 and we hope you will support that.

# Extending OMNI to an Embodied Kitchen Domain (AI2-Thor)

Some reviewers have asked for how OMNI might perform on more complex, open-ended domains, which are not easily solvable with simple heuristics or have an unlimited number of tasks. While the time window to add another domain has been small, because we too saw this challenge as the next frontier for OMNI, we have been doing our best each day since we submitted the original paper to try to get such experiments up and running. We have great news to report. We formulate OMNI to handle an infinite task space by continuously generating learnable and interesting tasks. Evaluated on a complex, visually realistic, 3D embodied kitchen robotics domain, AI2-THOR, OMNI significantly outperforms the controls. We have included results from this third, more complex, open-ended domain in the paper (Section 5).

In truly open-ended settings, there are an infinite number of possible tasks. To extend OMNI into this boundless task space, where the task set is not predetermined, we had to make a few new innovations. Essential to training an agent capable of handling any task in such an open-ended learning framework is the development of an universal reward function, that can evaluate if any task has been completed or not. To address this, the improved OMNI is not only able to endlessly generate and learn interesting tasks, but also generates code that enables the system to automatically verify if the task has been completed in this domain.

Applied to the AI2-THOR kitchen domain, OMNI demonstrates its effectiveness by continuously generating new tasks and significantly outperforming the uniform sampling baseline. Agents trained with OMNI learned 15 tasks (in only 1 million time steps), while those trained with uniform sampling learned none, highlighting OMNI's capability to facilitate learning in an infinite task space (and one that is complex and realistic, to boot). Section 5 of the revised manuscript contains more details and results of OMNI in the AI2-THOR domain.

Each reviewer can see replies to their specific questions in our reviewer-specific line-by-line responses below.

---

### Meta-Review · Area_Chair_65dF · 2023-12-09

**Metareview:**

This paper studies the problem of open-ended learning in reinforcement learning, where the goal is to enable an agent to learn how to solve novel and interesting tasks without supervision. The main challenge they tackle is how to quantify whether a task is interesting. To this end, the authors propose to use large language models (LLMs) to model interestingness.

The reviewers generally agreed that the paper is tackling an important problem, and that the proposed approach was novel and interesting. Given their recent progress, using LLMs to solve this task is a natural strategy, and while the proposed approach is somewhat simple, it appears effective at solving the problem. One concern is the reliance on the LLM encoding human priors about interestingness for certain domains, which may limit the generalizability of the approach to significantly new domains. There were also concerns about the environments and baselines. The new AI2-Thor environment helps address concerns that the environments in the evaluation are too simple.

**Justification For Why Not Higher Score:**

While the paper is tackling an important problem, the proposed approach is not significantly novel.

**Justification For Why Not Lower Score:**

The authors have convincingly demonstrated the effectiveness of their approach, including on the AI2-Thor environment.

---

### Decision · Program_Chairs · 2024-01-16

Accept (poster)